# Identification of enzymatic functions of osmo-regulated periplasmic glucan biosynthesis proteins from *Escherichia coli* reveals a novel glycoside hydrolase family

Sei Motouchi[1], Kaito Kobayashi [2], Hiroyuki Nakai[3] & Masahiro Nakajima [1✉]

Most Gram-negative bacteria synthesize osmo-regulated periplasmic glucans (OPG) in the periplasm or extracellular space. Pathogenicity of many pathogens is lost by knocking out *opgG*, an OPG-related gene indispensable for OPG synthesis. However, the biochemical functions of OpgG and OpgD, a paralog of OpgG, have not been elucidated. In this study, structural and functional analyses of OpgG and OpgD from *Escherichia coli* revealed that these proteins are β-1,2-glucanases with remarkably different activity from each other, establishing a new glycoside hydrolase family, GH186. Furthermore, a reaction mechanism with an unprecedentedly long proton transfer pathway among glycoside hydrolase families is proposed for OpgD. The conformation of the region that forms the reaction pathway differs noticeably between OpgG and OpgD, which explains the observed low activity of OpgG. The findings enhance our understanding of OPG biosynthesis and provide insights into functional diversity for this novel enzyme family.

[1] Department of Applied Biological Science, Faculty of Science and Technology, Tokyo University of Science, 2641 Yamazaki, Noda Chiba 278-8510, Japan. [2] Artificial Intelligence Research Center, National Institute of Advanced Industrial Science and Technology (AIST), 2-4-7 Aomi, Koto-ku, Tokyo 135-0064, Japan. [3] Faculty of Agriculture, Niigata University, 8050 Ikarashi 2-no-cho, Nishi-ku, Niigata 950-2181, Japan. ✉email: m-nakajima@rs.tus.ac.jp

Glycans play essential roles as energy sources and in forming structural components such as cell walls and exoskeletons. Glycans also facilitate interactions between organisms to enable processes such as pathogenicity, symbiosis, cell adhesion, and signaling. Osmo-regulated periplasmic glucans (OPG), whose carbohydrate moieties are composed of glucose, are synthesized in the periplasm or extracellular region by various Gram-negative bacteria. Although OPG were initially found as glycans synthesized at low osmolarity[1], various physiological roles of OPG, including pathogenic and symbiotic functions, have been reported. For example, pathogenicity of *Xanthomonas campestris*, *Agrobacterium tumefaciens,* and *Salmonella enterica* sv. *typhimurium*, regardless of phytopathogens or animal pathogens, is lost by knocking out OPG-associated genes[1,2]. Some species in Rhizobiaceae use OPG as a symbiotic factor[1,2]. Thus, OPG are important glycans, especially for enabling interactions between organisms.

OPG are classified into four groups based on their glucan backbone structures. Group 1 has a linear β-1,2-glucosyl backbone with β-1,6-glycosyl branches, whereas Groups 2–4 have cyclic backbones (cyclic β-1,2-glucan without side chains, cyclic β-1,2-glucan with one linkage substituted with α-1,6-linkage, and cyclic β-1,3-β-1,6-glucan, respectively)[1].

Groups 2 and 3 OPG are essentially β-1,2-glucans. Group 2 OPG are found in various species such as Rhizobiaceae and *Brucella* and are synthesized by cyclic glucan synthases (Cgs)[3–6] and transported to the periplasm by Cgt, an ABC transporter[7]. Group 3 OPG are found in *Ralstonia solanacearum, Xanthomonas campestris,* and *Rhodobacter sphaeroides*[8–10]. Because *R. sphaeroides* produces OPG even if the *cgs* gene is deleted, genes involved in synthesizing the β-1,2-glucan backbone are unclear[11]. In addition, the enzymatic mechanism that completes cyclization via an α-1,6-glucosidic linkage remains unknown. In contrast to these two groups, Group 4 OPG are formed with β-1,3- and β-1,6-glucosidic linkages by NdvB, which is composed of a glycosyltransferase family 2 domain and a glycoside hydrolase (GH) family 17 domain based on classification by the CAZy database[12] (http://www.cazy.org/), and NdvC, which is hypothesized to be a β-1,6-glucan synthase[13–16].

Group 1 OPG with linear backbones are found in various gamma proteobacteria such as *Escherichia coli, Pseudomonas aeruginosa* (an opportunistic pathogen), *Dickeya dadantii,* and *Pseudomonas syringae* (phytopathogens)[1]. Hereafter, the word OPG refers to Group 1 OPG. Genetic analysis indicates that the *opgGH* operon is responsible for OPG synthesis. This operon is distributed widely among almost all gamma proteobacteria[17]. Various pathogens lose their pathogenicity by knocking out *opgH* and/or *opgG* genes[1,2]. Recently, the OpgGH operon in *E. coli* was reported to be related to antibiotic resistance in silkworm[18]. In *D. dadantii* and *S. enterica* sv. *typhimurium*, OPG are known to regulate Rcs phosphorelay directly[2]. Rcs phosphorelay is a key system that regulates the expression of a set of genes encoding plant cell wall-degrading enzymes and the flhDC master operon, the flagellum structural gene of *D. dadantii*[2].

Chemical structures of OPG have been studied, especially in *E. coli*. Degrees of polymerization (DPs) of OPG are limited to 5–12[1]. OPG can be further modified with phosphoglycerol, succinate, and/or phosphoethanolamine groups by OpgB, OpgC, and/or OpgE, respectively[19–24]. In contrast to the physiological aspects and chemical structures of OPG, enzymes associated with synthesizing the glucosyl backbone of OPG have not been investigated fully. OpgH was suggested to synthesize a β-1,2-glucosyl main chain enzymatically[25]. However, these analyses were performed using a crude membrane fraction containing recombinant OpgH, and the reaction products were not confirmed biochemically to be β-1,2-glucans. OpgG is hypothesized

to play an important role in OPG backbone synthesis because an *opgG* knockout mutant cannot synthesize OPG, regardless of side chains[17]. Although it is postulated that OpgG is involved in forming β-1,6-glucosyl side chains, the ligand-free structure of OpgG from *E. coli* (EcOpgG) provided limited information on detailed biochemical functions[26]. OpgD from *E. coli* (EcOpgD), a paralog of EcOpgG with 32.9% amino acid sequence identity, is another key protein for linear OPG synthesis. Long linear β-1,2-glucosyl main chains with β-1,6-glucosyl side chains were detected in an *opgD* knockout mutant, suggesting that OpgD is related to adjusting chain lengths[27]. Furthermore, the Δ*opgD* mutant loses its motility and flagella, which are restored by deleting Rcs system-related genes[28]. However, no biochemical analysis of OpgD has been performed. Such a lack of biochemical evidence limits our understanding of the mechanism of OPG biosynthesis.

In this study, structural and functional analyses of EcOpgD and EcOpgG were performed. We discovered that EcOpgD is a β-1,2-glucanase (SGL, "S" is derived from sophoro-oligosaccharide, an alternative name of β-1,2-glucooligosaccharide) with a unique reaction mechanism, thereby establishing a new GH family (GH186) that is a phylogenetically new group, indicating that novel glycoside hydrolases are involved in OPG biosynthesis. Comparing the EcOpgD structure with EcOpgG with low SGL activity provided insights into the diversity of the functionally important region for hydrolysis by this GH family.

## Results

**Enzymatic properties of EcOpgD and EcOpgG.** Recombinant EcOpgD and EcOpgG fused with a His$_6$-tag at the C-terminus were produced using *E. coli* as a host and purified successfully. EcOpgD exhibited hydrolytic activity toward β-1,2-glucans to produce Sop$_{6-7}$ (Sop$_n$: sophoro-oligosaccharide with DP of n) preferentially, and Sop$_{8-10}$ accumulated during the late stage of the reaction (Fig. 1a left). Thus, β-1,2-glucans were used as substrates to investigate pH and temperature profiles. EcOpgD was stable at pH 4.0–10.0 and up to 50 °C. EcOpgD exhibited the highest activity at 40 °C and pH 5.0 (Supplementary Fig. 1a–d). These results imply that EcOpgG, a paralog of EcOpgD, also has SGL activity. EcOpgG was found to hydrolyze β-1,2-glucans; however, the catalytic velocity was very low (Fig. 1a right). Sop$_{6-8}$ were preferentially produced by EcOpgG, and Sop$_{9-11}$ also accumulated as final products. EcOpgG exhibited the highest activity at pH 5–7 (Supplementary Fig. 1e).

Among the various tested polysaccharides, EcOpgD exhibited high hydrolytic activity toward β-1,2-glucan but not other substrates, indicating that EcOpgD is highly specific to β-1,2-glucan (Fig. 1b top). Thus, kinetic analysis of EcOpgD using β-1,2-glucan as the substrate was performed (Fig. 1c left). The $k_{cat}$ of EcOpgD was comparable to those of SGLs from *Chitinophaga pinensis* (CpSGL) and *Talaromyces funiculosus* (TfSGL), the first sequence-verified SGLs found in a bacterium and a fungus, respectively (Table 1)[29,30]. The $K_m$ of EcOpgD was much higher than those of the two aforementioned SGLs, leading to a much lower $k_{cat}/K_m$ for EcOpgD. Nonetheless, the kinetic parameters of EcOpgD are still within the range of those of general GH enzymes. Endo-Dextranase from *Bacteroides thetaiotaomicron* shows a similar $K_m$ value toward dextran with a DP of approximately 2000 (estimated as 5.8 mg/mL)[31].

EcOpgG also showed the same substrate specificity toward the polysaccharides as that of EcOpgD. However, the reaction velocity of EcOpgG was low throughout the substrate concentration range examined (Fig. 1b bottom, c left). The relative activity against EcOpgD was only 0.79% at 8.0 mg/mL β-1,2-glucan (Table 1).

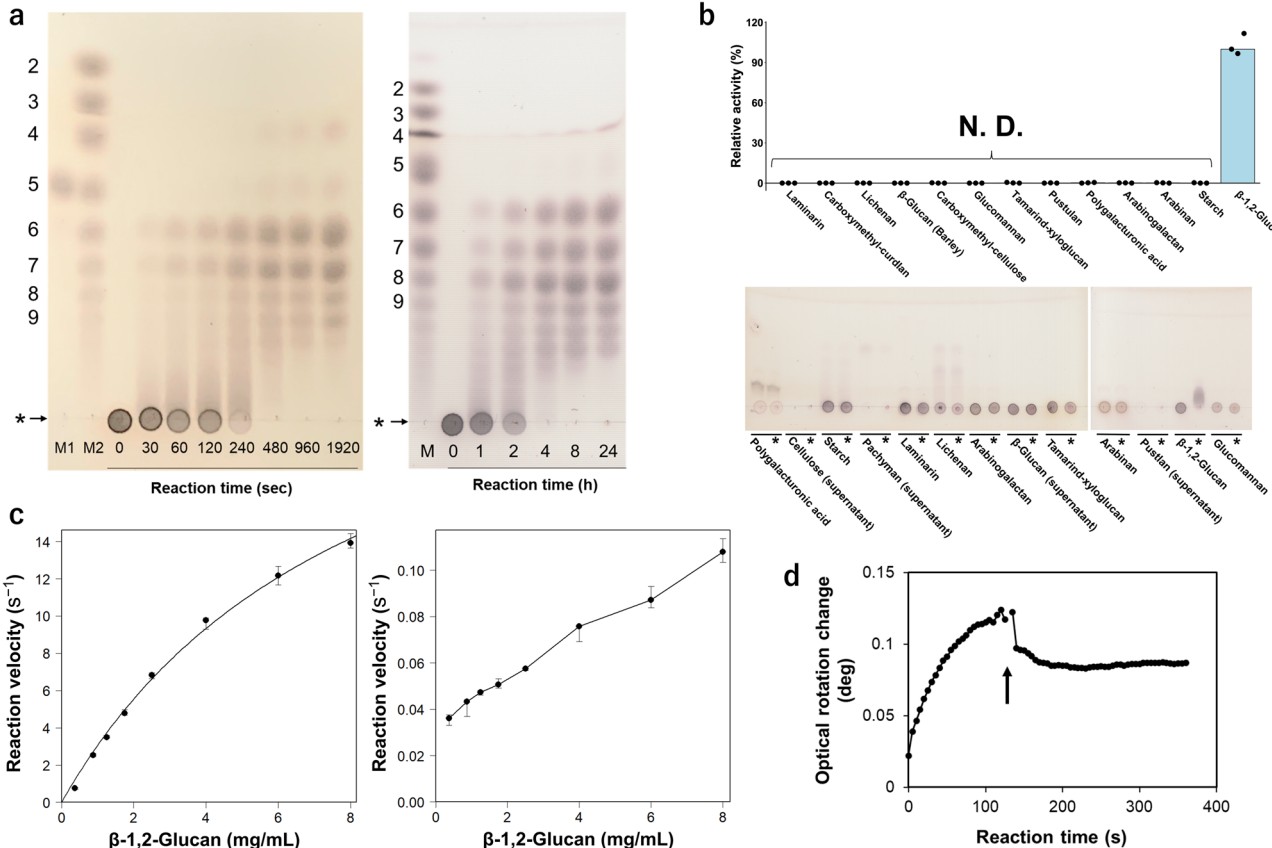

**Fig. 1 Catalytic analyses of EcOpgD and EcOpgG. a** TLC analysis of the action patterns of EcOpgD (left) and EcOpgG (right) on β-1,2-glucans. Lane M1, 10 mM Sop₅; Lanes M2 and M, a Sopₙs marker prepared using 1,2-β-oligoglucan phosphorylase from *Listeria inocua*. DPs of Sopₙs are shown on the left side of the TLC plates. Arrows represent β-1,2-glucan used for reactions. The origins of the TLC plates are shown as horizontal lines denoted by asterisks. EcOpgD and EcOpgG in the reactions are 0.14 mg/mL and 0.77 mg/mL, respectively. **b** Substrate specificity of EcOpgD (top) and EcOpgG (bottom). (top) N. D. represents values with less than 0.02% relative activity. Bars represent medians in triplicates. Triplicate data are plotted. (bottom) The reaction was performed at 37 °C for 24 h. Asterisks indicate that the reaction time was 24 h. Other lanes represent a reaction time of 0 h. **c** Kinetic analysis of EcOpgD (left) and EcOpgG (right). Data plotted as closed circles are medians in triplicates and the other data were used for error bars. (left) Data were regressed with the Michaelis–Menten equation (solid line). (right) Plots were medians in triplicates and simply connected by lines because of poor fitting to the Michaelis–Menten equation. **d** Time course of the observed optical rotation during β-1,2-glucan-hydrolysis by EcOpgD. The arrow indicates that several drops of aqueous ammonia were added to the reaction mixture 120 s after the reaction started.

**Table 1 Kinetic parameters of EcOpgD for β-1,2-glucans and comparison with known SGLs.**

| Enzyme | $K_m$ (mg/mL) | $k_{cat}$ (s⁻¹) | $k_{cat}/K_m$ (s⁻¹ mg⁻¹ mL) |
|---|---|---|---|
| EcOpgD | 8.6 ± 1.1 (0.44 ± 0.056)[b] | 29 ± 2 | 3.4 ± 0.2 |
| CpSGL[a] (GH144) | 0.71 ± 0.15 (0.068 ± 0.014)[b] | 49 ± 4 | 69 ± 10 |
| TfSGL[a] (GH162) | 0.18 ± 0.02 (0.015 ± 0.001)[b] | 31 ± 1 | 170 ± 10 |
| EcOpgG | | (0.11)[c] | |

[a]The values of CpSGL and TfSGL are cited from Abe et al.[29] and Tanaka et al.[30], respectively.
[b]The values calculated using molar concentrations (mM) are shown in parentheses.
[c]Reaction velocity in the presence of 8 mg/mL β-1,2-glucan at pH 5.5 is shown in parenthesis.

The reaction mechanisms of glycoside hydrolases are basically divided into two types: anomer-retaining and -inverting types. An anomer orientation of a glycoside moiety at reducing end of a substrate is retained after the reaction in the former type, while inverted in the latter type. Determination of which mechanism the enzyme follows is the first step to dissect a detailed reaction mechanism. Therefore, we measured optical rotation of EcOpgD during the reaction in the same way as the reaction mechanisms of GH144 and GH162 enzymes were determined[29,30] (see "Methods" to read the principle of the experiment). In EcOpgD, an increase in the degree of optical rotation during the early stage

of the reaction and a rapid decrease in the value upon adding aqueous ammonia were observed (Fig. 1d). This result is similar to observations made for CpSGL and TfSGL, which are anomer-inverting SGLs[29,30]. Thus, EcOpgD follows the anomer-inverting mechanism.

**Overall structure of ligand-free EcOpgD.** The ligand-free structure of EcOpgD was determined at 2.95 Å resolution (Fig. 2a left). A dimer was used for describing the EcOpgD structure because the stable assembly in the structure was found to be a dimer by calculation using PISA server (https://www.ebi.

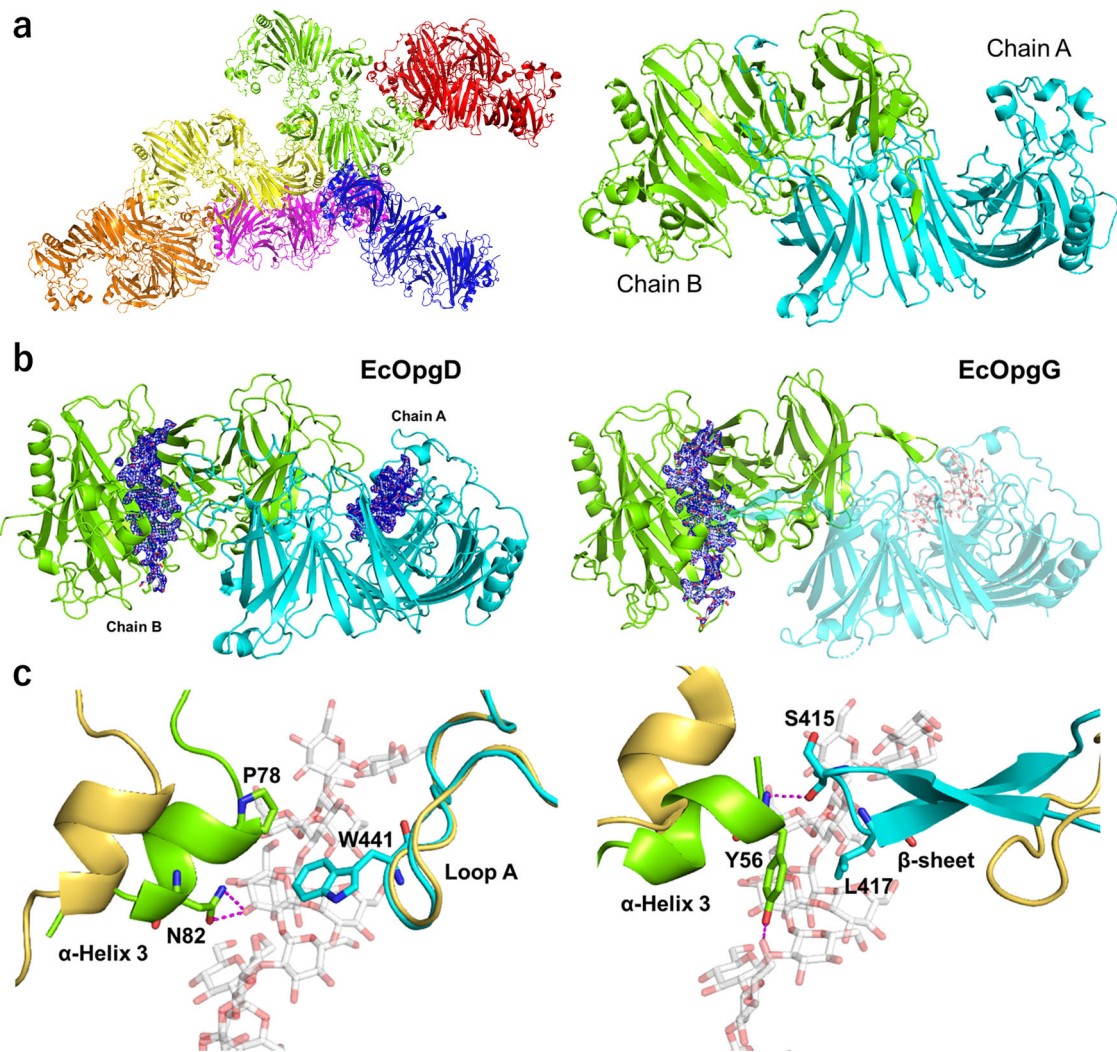

**Fig. 2 Structures of EcOpgD and EcOpgG. a** Overall structure of ligand-free EcOpgD. (left) Structure of an asymmetric unit. There are 12 monomers in an asymmetric unit with almost identical conformations (RMSD within 0.30 Å). The free energy of assembly dissociation ($\Delta G_{diss}$) was calculated by PISA[32]. All $\Delta G_{diss}$ values of dimer interfaces in the asymmetric unit are higher than 32.9 kcal/mol, indicating that the dimer is a stable assembly. Thus, the six dimers are shown in red, blue, magenta, light green, yellow, and orange. (right) Biological assembly. Chains A and B are shown in cyan and light green, respectively. The RMSD between EcOpgD and EcOpgG (PDB: 1txk) is 2.206 Å. **b** Overall structures of Michaelis complexes of the EcOpgD D388N (left) and EcOpgG D361N (right) mutants. Chains A and B are shown in cyan and light green, respectively. The two subunits represent a biological assembly. The electron densities of β-1,2-glucans are shown as $F_o$–$F_c$ omit maps by blue meshes at the 3σ contour level. Substrates are shown as white sticks. (left) The biological assembly is identical with molecules in an asymmetric unit (RSMD: 0.170 Å). (right) A plausible bioassembly of EcOpgG in solution is a dimer, according to PISA analysis[32]. Chains A and B are a symmetry mate. **c** Superposition between the ligand-free and the Michaelis complex structure of EcOpgD (left) and EcOpgG (right) around the Loop A region. The ligand-free structures are shown in light yellow. (left) P78, N82, and W441 in the complex structure are shown as sticks. (right) Y56, S415, and L417 in the complex structure are shown as sticks.

ac.uk/pdbe/pisa/)[32] and size-exclusion chromatography (SEC) (Fig. 2a right, Supplementary Fig. 2). In addition, ligand-free EcOpgG was also found to assemble as a dimer by SEC in the presence of NaCl (Supplementary Fig. 2). This result is discussed in Supplementary Note. The overall structure of the ligand-free EcOpgD resembles that of the ligand-free EcOpgG. The RMSD between EcOpgD and EcOpgG (PDB ID: 1txk) is 2.206 Å (Supplementary Fig. 3).

**Michaelis complexes of EcOpgD and EcOpgG.** The D388N mutant was used to determine a Michaelis complex structure of EcOpgD because this mutant displayed the lowest activity among mutants of acidic amino acids conserved in its homologs, and this residue is located in the long cleft of the ligand-free structure

(Table 2, Supplementary Fig. 4). The complex structure was obtained as a dimer by co-crystallizing the D388N mutant with β-1,2-glucan (Fig. 2b left). In the catalytic pockets of chains A and B, the electron densities of β-1,2-glucans with DP11 and DP13 ($Sop_{11}$ and $Sop_{13}$, respectively) were clearly observed, respectively (Fig. 2b left, Supplementary Fig. 5a). Chain B was used for describing the complex because the glucan chain is longer.

The substrate binding site is formed as a cleft in the ligand-free structure, whereas this site forms a tunnel in the structure of the complex (Fig. 2a, b, Supplementary Fig. 6). The difference arises mainly from the closure motion of α-helix 3 to the substrate in the cleft. In the helix of the complex structure, P68 is located in close proximity to W441, and N82 forms hydrogen bonds with the 3-hydroxy group of the Glc moiety (Fig. 2c left). However, substituting P78 (P78A and P78L) and N82 (N82A) only had a

**Table 2 Specific activities of EcOpgD mutants.**

|  | Specific activity (U/mg) | Relative activity (%) |
|---|---|---|
| E209Q | 0.48 (0.028) | 3.5 |
| D300N | 0.13 (0.0048) | 0.90 |
| D351N | 3.3 (0.31) | 23 |
| E385Q | 0.28 (0.014) | 2.0 |
| D388N | 0.022 (0.0009) | 0.16 |
| E439Q | 0.77 (0.018) | 5.6 |
| E445Q | 1.9 (0.073) | 14 |
| R359A | 0.064 (0.0013) | 0.46 |
| T386L | 0.78 (0.024) | 5.6 |
| T386A | 0.80 (0.013) | 5.8 |
| T386S | 2.4 (0.059) | 17.6 |
| W441L | 0.87 (0.013) | 6.3 |
| W441F | 1.0 (0.013) | 7.3 |
| P78L | 8.2 (0.50) | 59 |
| P78A | 6.5 (0.064) | 47 |
| N82A | 12 (0.20) | 87 |
| Y356F | 2.2 (0.054) | 16 |
| Wild-type | 14 (0.33) | 100 |

Medians of triplicate experiments are presented.
Maximum absolute values between medians and the other data of the triplicate experiments are shown in parentheses.
Activity to produce 1 μmol of reaction product per minute is defined as 1 U (μmol/min).

mild effect on activity (Table 2), implying that the closure motion of α-helix 3 is not important for catalysis. Unlike the motion of α-helix 3, chain A is involved in the reaction mechanism. In particular, residues 434–453, hereafter termed Loop A, are important for catalysis, as described below.

To generate hypotheses regarding the very low reaction velocities of EcOpgG, we determined the complex structure of EcOpgG (D361N mutant) with β-1,2-glucan at 1.81 Å resolution (Fig. 2b right). In the catalytic pocket of the complex, the electron density of a $Sop_{16}$ molecule was observed (Supplementary Fig. 5b). The dynamic conformational change from a loop (residues 409–425) to two β-strands upon substrate binding is a unique feature found in EcOpgG (Fig. 2c right). However, the β-sheet interacts not with the substrate but with the α-helix 3 in the complex. The motion of α-helix 3 is induced by the interactions between Y56 in the helix and the 6-hydroxy group of the Glc moiety. These observations suggest that the closure motion of α-helix 3 is required for forming the β-sheet. In contrast, Loop A in EcOpgD does not change its conformation upon substrate binding nor does it have electrostatic interactions with α-helix 3 (Fig. 2c left). Because α-helix 3 in EcOpgD is not required for catalytic activity (see above), the motion of this helix may be a remnant of molecular evolution from EcOpgG.

**Substrate binding mode of EcOpgD and EcOpgG.** We searched for distorted Glc moieties in EcOpgD to determine the position of the glycosidic bond cleaved. Subsite −1 is often distorted to enable a nucleophile to attack an anomeric center. For example, the Glc moieties at subsite −1 of TfSGL (GH162)[30] and Lin1840 (GH3)[33] form skew boat ($^1S_3$) conformations which are distorted conformations. In EcOpgD, the seventh and eleventh Glc moieties from the reducing end of $Sop_{13}$ form skew boat ($^1S_3$ and $^1S_5$, respectively) conformations (Supplementary Fig. 7a), according to the puckering coordinate system of a six-membered pyranose ring, Cremer–Pople parameters[34]. The other Glc moieties form a chair ($^4C_1$) conformation. A water molecule (Wat1) is located near the anomeric carbon of the seventh Glc moiety from the reducing end (3.5 Å). The angle formed by this water, the anomeric carbon and the glycosidic bond oxygen atom is 168.17°, which is close to 180° and suitable for the nucleophilic (in-line)

attack of the anomeric carbon (Fig. 3a). The position of the water molecule is consistent with the result that EcOpgD follows an anomer inverting mechanism. The other Glc moiety with the skew boat conformation has no nucleophilic water because the O5 atom of the twelfth Glc moiety from the reducing end occupies the position for a nucleophile (Supplementary Fig. 7b). These observations strongly suggest that the position of the seventh Glc moiety is subsite −1 and that EcOpgD accommodates a $Sop_{13}$ molecule at subsites −7 to +6 as a Michaelis complex (Supplementary Fig. 8a). This is consistent with the preferential production of $Sop_{6-7}$ by hydrolysis of β-1,2-glucans (Fig. 1a left).

At least two residues participate in the interaction with each Glc moiety at subsites −5 to +4. In particular, the Glc moiety at subsite −1 is recognized by five residues, which allows the distorted conformation. The Glc moieties at subsites −6, +5, and +6 are mainly recognized intra-molecularly rather than forming hydrogen bonds with nearby residues (Supplementary Table 1, Supplementary Fig. 8a).

A β-1,2-glucan molecule binds to the substrate pocket in the complex of EcOpgG at almost the same position as EcOpgD, except at the subsites −5 to −7 in EcOpgD. Substrate recognition residues are well conserved at subsites −1 to +4 between the two enzymes but are not at other subsites, especially on the minus side (Supplementary Table 1, Supplementary Fig. 8b). Although the conformation of the Glc moiety at subsite −5 in EcOpgD is $^1S_5$, the corresponding Glc moiety in EcOpgG forms the $^4C_1$ conformation. The only Glc moiety forming $^1S_3$ in EcOpgG is the well-superimposed Glc moiety at subsite −1 in EcOpgD. However, electron density for nucleophilic water is poor in EcOpgG, which strongly supports the observed low SGL activity of EcOpgG (Fig. 3b).

**Catalytic residues of EcOpgD.** Generally, GH enzymes have two acidic residues as catalysts. D388 was identified as a general acid catalyst candidate because this acidic amino acid directly interacts with the oxygen atom of the glycosidic bond between subsites −1 and +1 (Fig. 3a). The distance between the nitrogen atom of the N388 side chain and O2 atom is 3.1 Å, and the dihedral angle for O5′–C1′–O2–C2 is within the range of antiperiplanar (−158.4°) (Fig. 3c), suggesting that the substrate and D388 are properly arranged for *syn* protonation[35,36]. The relative activity of D388N was the lowest in all examined mutants (0.15%) (Table 2). In addition, CD spectra of all mutant enzymes whose structures have not been solved were analyzed, resulting in almost the same patterns (Supplementary Fig. 9). These results strongly suggest that D388 is a general acid residue.

No acidic residue directly interacting with the nucleophilic water is found in the complex structure, unlike canonical anomer inverting GH enzymes. Thus, proton dissociable amino acid residues (D300, Y356, R359, and E385) on possible proton transfer pathways from the nucleophilic water were listed as general acid candidates (Fig. 3a). All listed residues need at least three water molecules, including the nucleophilic water, in order to transfer the proton from a nucleophilic water to each residue. Hereafter, these water molecules are called Wat1, Wat2, and Wat3 from the nucleophilic water. Although Wat2 interacts with no proton dissociative residues, Wat3 interacts directly with Y356. The relative activity of Y356F was not fully decreased (14%) (Table 2), suggesting that Y356 is not a general base. This result also suggests that E385 beyond Y356 is not a general base despite the dramatic decrease in activity observed for the E385Q mutant (Fig. 3a, Table 2). D300 and R359 interact with Wat3 via the 4-hydroxy group in the Glc moiety at subsite −1. The relative activities of D300N and R359A were 0.89% and 0.47%, respectively (Table 2). The

predicted p$K_a$ values by propKa 3.4.0[37] for the carboxy group of D300 and guanidium group of R359 are 4.1 and 13.8, respectively. An electrostatic interaction between D300 and R359 probably promotes deprotonation of the D300 carboxy group as a general base and protonation of the guanidium group of R359. The p$K_a$ value of the D388 carboxy group substituted with asparagine in the complex structure of the D388N mutant was predicted to be 5.5, which is consistent with the optimum pH 5.0 between the predicted p$K_a$ values of the two carboxy groups in D300 and D388. Overall, a nucleophilic water is likely activated by D300 as a general base through the two water molecules (Wat2 and Wat3) and one substrate hydroxy group.

**Environment around the water molecules on the proposed catalytic pathway.** The high efficiency of proton relays that carry a proton from a nucleophilic water to a general base via other water molecules has been explained using the Grotthuss mechanism in GH enzymes (GH6 and GH136)[38,39]. To further judge the plausibility of the suggested proton transfer pathway, the environment around Wat1 (nucleophilic water), Wat2 and Wat3 was observed, and mutational analysis was performed. The

three water molecules are structurally sequestered from the solvent, especially by the β-1,2-glucan and Loop A (Fig. 4). Other loops (residues 349–355 and 380–389 of chain B) also participate in sequestering these water molecules from the solvent.

The nucleophilic water is surrounded by W441 and T386 (Fig. 4a, b). Substitution of W441 with Phe or Leu, smaller hydrophobic side chain amino acids, resulted in greatly reduced relative activities (7.7% and 6.5%, respectively) (Table 2). Interaction of the hydroxy group of T386 with E445 causes the methyl group of T386 to face the nucleophilic water. The relative activity of T386S, a mutant without a methyl group, decreased (17.5% relative activity) (Table 2). These results suggest that the hydrophobicity provided by the six-membered ring moiety of W441 and the methyl group of T386 around the nucleophilic water support efficient proton transfer by the Grotthuss mechanism. Wat2 is surrounded by the Cβ of E385, C5 of the Glc moiety at subsite −1 and the main chain oxygen atom of G440 (Fig. 4a, b). The G440 oxygen atom forms a hydrogen bond with Wat2, suggesting that G440 fixes Wat2 to an appropriate position for intermediate proton transfer from Wat1 to Wat3. Wat3 is sequestered from the solvent by the 6-hydroxy group of the Glc moiety at subsite −3, the 4-hydroxy group of the Glc moiety at subsite −1, Y356 and R359 (Fig. 4a, c). Wat3 forms hydrogen

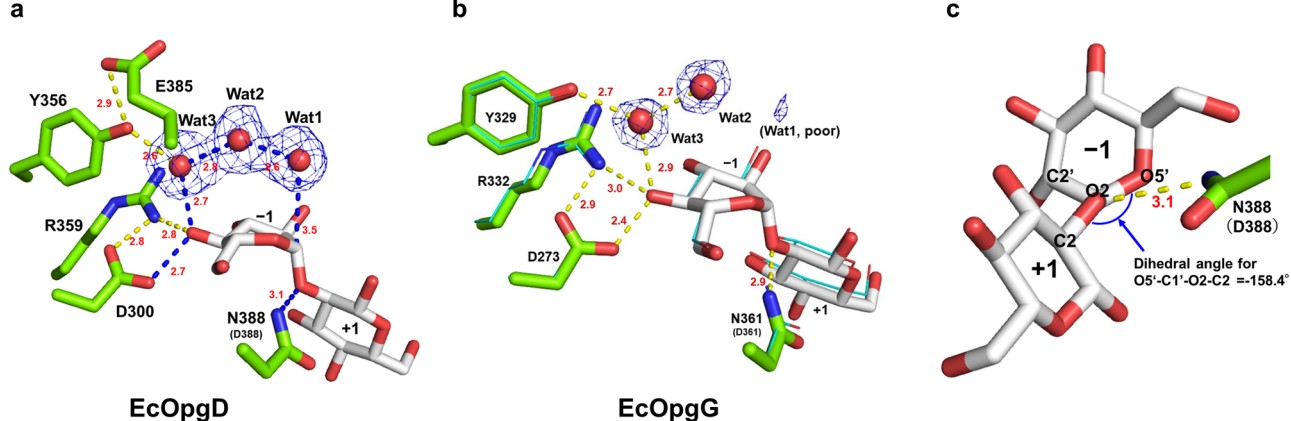

**Fig. 3 Substrate binding modes of EcOpgD and EcOpgG complex structures.** Hydrogen bonds with distances (Å, red numbers) are shown as yellow dotted lines. **a, b** Structures around subsite −1 in EcOpgD (**a**) and EcOpgG (**b**). Hydrogen bonds on the suggested reaction pathway are shown as blue dotted lines. The electron densities of Wat1, Wat2, and Wat3 are shown as $F_o$–$F_c$ omit maps by blue meshes at the 3σ contour level. Residues and substrates are shown in green and white sticks, respectively. **b** EcOpgD (cyan, line representation) superimposed with EcOpgG. **c** Conformation of the Glc moieties at subsites −1 and +1 in EcOpgD. The innate wild-type residues are shown in parentheses if the residues are substituted.

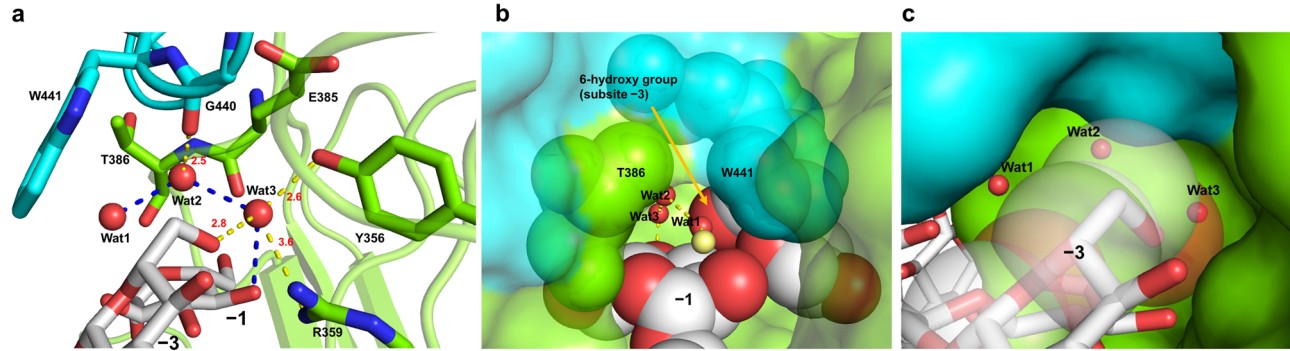

**Fig. 4 Water molecules indispensable for the reaction pathway.** Chains A and B are shown in cyan and light green, respectively. Hydrogen bonds with distances (Å, red numbers) are shown as yellow dotted lines. Substrates are shown as white sticks. **a** Residues sequestering water molecules on the reaction pathway. All residues used for fixing Wat1–3 on the reaction pathway are shown as sticks. Blue dotted lines represent a route from the nucleophile to a substrate hydroxy group on the reaction pathway. **b, c** The environment around the sequestered Wat1 (**b**) and Wat2 and Wat3 (**c**). **b** The substrate and T386 and W441 are shown as spheres with van der Waals radii. **c** The substrate is shown in stick representation. C1, C5, O5, and O6 atoms of the Glc moiety at subsite −3 are shown as semi-translucent spheres with van der Waals radii.

bonds with the 6-hydroxy group of the Glc moiety at subsite −3 and the hydroxy group of Y356. The reduced activity of Y356F (16% relative activity) suggests that Y356 is not a general base, as described above, but its hydroxy group is important for fixing and orienting Wat3 (Table 2). Y356 is not a general base, as described above, but its hydroxy group is important for fixing and orienting Wat3 (Table 2).

Loop A, important for sequestering water molecules, is tethered by interactions involving E385, E439, and E445 (Supplementary Fig. 10). The relative activities of the E385Q, E439Q, and E445Q mutants were greatly reduced (2.0%, 5.6%, 14%, respectively) (Table 2). Even substituting glutamate residues with similar amino acids affected hydrolytic activity noticeably, indicating the vital role of Loop A in the catalytic mechanism of EcOpgD. Overall, structural data and mutational analyses support the suggested proton transfer pathway. We summarize the proposed reaction mechanism of EcOpgD in Fig. 5.

**Comparison of the catalytic mechanism with EcOpgG.** Intriguingly, residues associated with catalysis in EcOpgD (D388, R359, D300) are conserved in EcOpgG (Fig. 3a, b, Supplementary Figs. 4, 8c), although the reaction velocity of EcOpgG was very low (Fig. 1c right). In addition, the binding free energies between Sop$_{13}$ and the enzymes (EcOpgD and EcOpgG) were calculated by using scoring function of smina software[40], resulting in almost the same $\Delta G$ values (−6.34 kcal/mol and −5.87 kcal/mol, respectively). These results indicate significant differences not in binding affinity for substrates but in the environments around the reaction route between EcOpgG and EcOpgD. In fact, a poor nucleophilic water was observed in EcOpgG, as described above (Fig. 3b). This difference in activity appears to be caused by the hydrophobic amino acids surrounding the nucleophilic water. W441 in EcOpgD is spatially substituted with L417 in EcOpgG (Figs. 2c, 6), whereas T386 in EcOpgD is conserved in EcOpgG (Supplementary Figs. 4, 8c). The other difference is the environment around Wat2. In EcOpgG, two water molecules form a route to facilitate the escape of Wat2 to solvent, whereas G440 in EcOpgD blocks this route completely (Fig. 6). These observations explain the low activity of EcOpgG toward β-1,2-glucans and strongly support the suggested proton transfer pathway of EcOpgD.

## Discussion

EcOpgG and EcOpgD have been classified at the genetic level as proteins indispensable for OPG biosynthesis and regulation of OPG chain lengths, respectively[17,27]. Although the shape of the ligand-free EcOpgG cleft suggests binding to glycans[26], the enzymatic functions of EcOpgD and EcOpgG have not been investigated. In this study, EcOpgD is shown to be an SGL (β-1,2-glucanase), strongly suggesting that EcOpgD can adjust OPG chain lengths enzymatically. There are sufficient spaces to accommodate β-1,6-glucosyl side chains at subsites +3, +5, and +6, suggesting that EcOpgD can interact with β-1,6-branched β-1,2-glucans as substrates (Supplementary Fig. 11). In addition, EcOpgG has sufficient spaces at the same subsites as EcOpgD, and additionally at subsites −9, −8, −7, and −5 (Supplementary Fig. 11).

EcOpgG displayed much lower SGL activity than EcOpgD. However, the catalytic residues and R359 supporting deprotonation of D300 are highly conserved in OpgG and OpgD homologs. Moreover, the distorted Glc in the Michaelis complex of EcOpgG structurally resembles that of EcOpgD at subsite −1 (Fig. 3b), suggesting that β-1,2-glucan can bind to EcOpgG with a suitable conformation for hydrolysis. However, the absence of a nucleophilic water molecule and the presence of a branched proton transfer pathway from Wat2 may be responsible for the low specific activity of EcOpgG (0.11 U/mg). Overall, EcOpgG is intrinsically a GH (glycoside hydrolase) enzyme but may require a cofactor to display greater specific activity.

The possibility that EcOpgG is in fact a transglycosylase of the β-1,6-glucose side chain remains open. Knockout mutants of OpgG are supposed not to synthesize OPG precursors with β-1,2-glucan main chain[1]. However, this may simply be because the precursors are not detected. *E. coli* has only one β-glucosidase homolog predicted to be localized in the periplasmic space. The β-glucosidase from *E. coli* is a close homolog of β-glucosidases specific to Sop$_n$s as native substrates[33,41], and the recognition residues for subsite +1 in the homologs are conserved in the *E. coli* β-glucosidase. Therefore, if knockout of *opgG* results in just synthesizing unbranched β-1,2-glucans, they are likely to be degraded.

OpgH has often been predicted to be a cofactor of OpgG because *opgG* and *opgH* form a gene cluster[26]. However, *opgI* and *opgH* from *R. sphaeroides* complement Δ*opgH E. coli* even though the sequences of the periplasmic region are diverse among OpgH homologs[11], which suggests that OpgG has a periplasmic cofactor other than OpgH.

According to the annotation by InterPro[42], EcOpgG and EcOpgD belong to the MdoG superfamily, which comprises the MdoD and MdoG families (MdoG and MdoD are the former names of OpgG and OpgD, respectively). Although this superfamily can be divided into 14 clades (Supplementary Fig. 12), members of both families are mixed phylogenetically based on the annotation, which generates a degree of confusion. Highly conserved residues among the MdoG superfamily are located in the substrate pocket (Supplementary Fig. 13a), which contains the catalytic residues identified in this study (Supplementary Fig. 13b). Many of the conserved residues contribute strongly to binding of a β-1,2-glucan molecule, according to estimates of the Gibbs free energy for binding (Supplementary Fig. 13c). In contrast, the remarkable difference in biochemical functions between EcOpgG and EcOpgD was attributed to a region of Loop A.

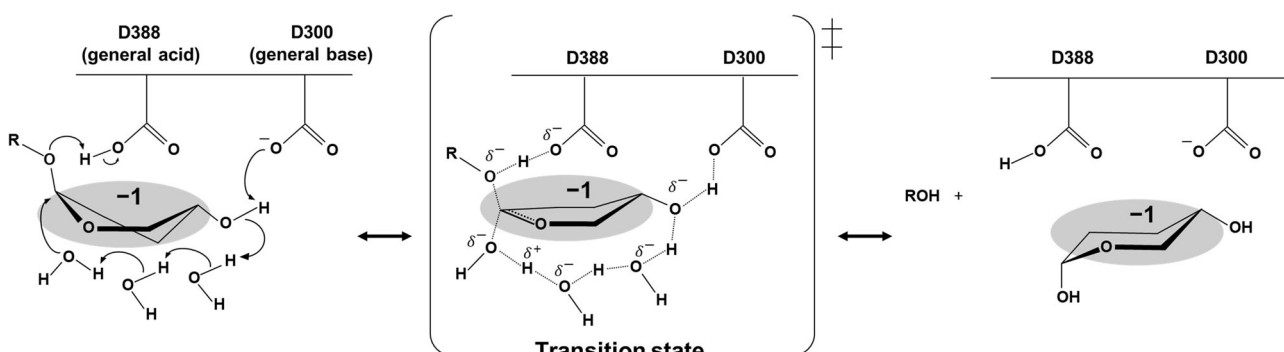

**Fig. 5 A proposed reaction mechanism of EcOpgD.** The Glc moiety at subsite −1 is highlighted in gray. Arrows represent the pathway for electron transfer.

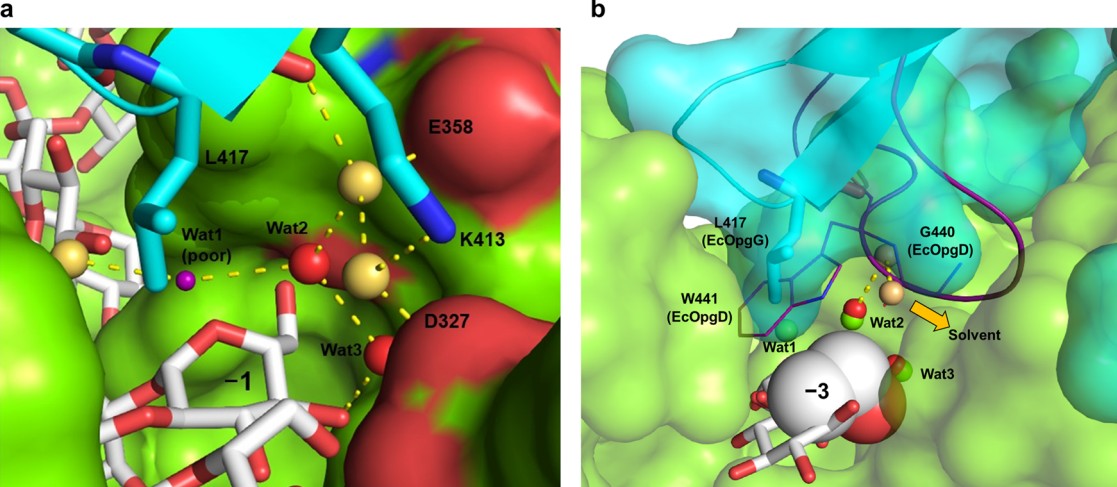

**Fig. 6 The environment around Wat2 and Wat3 in the EcOpgG complex structure. a** The positions of the water molecules around subsite −1 of EcOpgG. A small purple sphere is shown as a tentative water molecule with poor electron density (Wat1). Wat2 and Wat3 and the other water molecules are shown in red and beige spheres, respectively. K413 and L417 are shown as cyan sticks. Chains A and B are shown in blue cartoon and green surface representations, respectively. The substrate is shown as a white stick. **b** Superposition showing the spatial position of Loop A in EcOpgG and EcOpgD. The superimposed Loop A in the EcOpgD complex is shown in purple. G440 and W441 are shown as lines. Wat1–3 molecules in EcOpgD are shown as light green spheres. EcOpgG is shown as a semi-translucent surface. L417 is shown in cyan. The water molecules of EcOpgG are shown as presented in (**a**), except that Wat1 (poor) is omitted. The Glc moieties at subsites −1 and −3 of EcOpgG are shown as white sticks. C5, C6, and O6 atoms of the Glc moiety at subsite −3 in EcOpgG are shown as spheres of van der Waals radii. The thick yellow arrow represents another proton network that leads to the solvent.

In particular, W441 is important for the fixation of nucleophilic water in EcOpgD; however, this amino acid and G440 are conserved in a limited group, clade 2, containing EcOpgD (Supplementary Fig. 13d). In EcOpgG, L417 is spatially equivalent to W441 in EcOpgD and is conserved only in close homologs of EcOpgG (clade 14). Comparing the Loop A region between clades, the GXGG motif shared among clades 1–4 may represent a functionally common structure (Supplementary Fig. 13d). No other consensus region or residue was identified. Furthermore, some clades have no region corresponding to Loop A, indicating diverse biochemical functions for this superfamily. Knockout of OPG-related genes drastically alters phenotypes related to pathogenicity in many species[1,2]. Homologs from such species in the MdoG superfamily are found only in clades 2–4 and 14, whereas there is no study on the functions of homologs in clades 5–13. Our study on the structure–function relationships of EcOpgD and EcOpgG will provide important clues to investigate relationships between the enzymatic functions of OPG-related genes and the phenotypes of these species.

GHs are classified into families in the CAZy database based on their amino acid sequences[12] (http://www.cazy.org/). To understand the category of the MdoG superfamily, we initially compared EcOpgD with the GH144 and GH162 family SGLs, which are all known SGLs. EcOpgD showed neither amino acid sequence similarity nor tertiary structure similarity with GH144 and GH162 SGLs[29,30]. Performing a BLAST search using EcOpgD as a query against the database containing all GH families gave no GH family enzyme hit with sufficient amino acid sequence identity. According to a DALI (http://ekhidna. biocenter.helsinki.fi/dali_server/)[43] search, GH38 α-mannosidase (PDB: 2wyh) was the top hit among GH family enzymes. However, only the C-terminal domain, but not the catalytic domain of the GH38 enzyme, showed structural similarity with EcOpgD. Therefore, the MdoG superfamily represents a new GH family, GH186.

We also proposed a unique reaction mechanism of EcOpgD (Fig. 5). The proton transfer pathway from a nucleophilic water to a general base is unprecedently long among GH families,

involving two water molecules and a substrate hydroxy group. No pathway to deprotonate nucleophilic water by a general base via a substrate like EcOpgD has been found in GH families. The number of water molecules passing through is also unprecedented among GH families. Nevertheless, the reaction model is credible because each feature is found in known reaction mechanisms. In GH130 and GH162, protonation of a scissile bond oxygen atom is carried out by a general acid via a substrate hydroxy group[30,44,45]. In GH6, GH101, GH136, and GH162[30,38,39,46], only one water molecule is inserted in the proton transfer pathway from a nucleophilic water to a general base. The findings in this study expand the diversity of GH enzymes in terms of reaction mechanisms and phylogenetic groups.

We identified the enzymatic function of EcOpgD and suggested a unique reaction mechanism of the enzyme. In contrast, EcOpgG was found to show low SGL activity but have a substrate binding mode suitable for hydrolysis. Intriguingly, comparing these two enzymes revealed that amino acid sequences in the Loop A region are not conserved in the MdoG superfamily, suggesting diverse reaction mechanisms for this family. Discovery of novel GH enzymes involved in OPG biosynthesis and elucidation of reaction mechanisms based on three-dimensional structures underpin future efforts that will focus on understanding OPG biosynthesis. This study should lead to unveiling more interactions between organisms through OPG and facilitate efforts to control OPG biosynthesis in Gram-negative bacteria for regulating pathogenicity and symbiosis.

## Methods

**Cloning and purification of EcOpgD and EcOpgG**. Genes encoding EcOpgG (GenBank: CAQ31569.1) and EcOpgD (GenBank: CAQ31910.2) were amplified by PCR with primer pairs shown in Supplementary Table 2 using PrimeSTAR Max (Takara Bio, Shiga, Japan) and a colony of *E. coli* BL21(DE3) as the template. The forward primers were designed to eliminate the signal sequence predicted by the SignalP5.0 server[47]. The amplified genes were inserted between the XhoI and NdeI sites of the pET30a vector by the SLiCE method[48] to add a

C-terminal His$_6$-tag to the target proteins. The constructed plasmids were transformed into *E. coli* BL21(DE3), and the transformants were cultured in 1 L Luria-Bertani medium containing 30 mg/L kanamycin at 37 °C until the absorbance at 600 nm reached 0.6. Expression was then induced by adding isopropyl β-D-1-thiogalactopyranoside to a final concentration of 0.1 mM, and cells were cultured at 20 °C (EcOpgD) or 7 °C (EcOpgG) for a further 24 h. The cells were centrifuged at 7000 × *g* for 10 min and suspended in 50 mM Tris-HCl buffer (pH 7.5) containing 50 mM NaCl (buffer C). The suspended cells were disrupted by sonication and centrifuged at 33,000 × *g* for 15 min to obtain a cell extract. The cell extract was applied onto a HisTrap™ FF crude column (5 ml; Cytiva, MA, USA) pre-equilibrated with a buffer containing 50 mM Tris-HCl (pH 7.5), 500 mM NaCl and 20 mM imidazole. After the column had been washed with the same buffer, the target protein was eluted with a linear gradient of 20–300 mM imidazole in a buffer containing 50 mM Tris-HCl (pH 7.5) and 500 mM NaCl. Amicon Ultra 30,000 molecular weight cutoff centrifugal filters (Millipore, MA, USA) were used to concentrate a portion of the fractionated protein and to exchange the buffer to 50 mM Tris-HCl (pH 7.5) containing 50 mM NaCl. Each purified protein migrated as a single band of 60 kDa on SDS-PAGE gels, which is consistent with the theoretical molecular mass of EcOpgD (60601 Da) or EcOpgG (56543 Da). Concentrations of purified enzymes were calculated from absorbance at 280 nm[49].

**β-1,2-Glucans used for experiments.** β-1,2-Glucans with an average DP of 121 calculated from their number average molecular weight (Mn) were treated with NaBH$_4$ for the MBTH (3-methyl-2-benzothiazolinonehydrazone) assay[50]. The average DP of β-1,2-glucans used for TLC analysis is 121 based on Mn. The average DP of β-1,2-glucans used for crystallization to obtain the Michaelis complexes of EcOpgD and EcOpgG were 121 and 17.7 based on Mn, respectively. The average DP of β-1,2-glucans used for polarimetric analysis was estimated to be approximately 25, according to the preparation method[51]. All β-1,2-glucans used in this study were prepared by ourselves.

**MBTH method.** Quantification of reaction products released by enzymatic reactions was performed by the MBTH method[52]. After each reaction, 20 μL aliquots of the reaction mixtures were taken and heated at 100 °C for 5 min. The samples were mixed with 20 μL of 0.5 N NaOH and then with 20 μL MBTH solution composed of 3 mg/mL MBTH and 1 mg/mL dithiothreitol. The mixtures were incubated at 80 °C for 30 min. A solution comprising 0.5% FeNH$_4$(SO$_4$)$_2$, 0.5% H$_3$NSO$_3$, and 0.25 N HCl (40 μL) was added to the mixtures, and then 100 μL distilled water was also added after cooling to room temperature. The absorbance at 620 nm was measured. Sop$_2$ (0.3–2.0 mM) was used as the standard.

**TLC analysis.** EcOpgD (0.14 mg/mL) and EcOpgG (0.77 mg/mL) were incubated with 1% β-1,2-glucan in 25 mM sodium acetate buffer (pH 5.0) and sodium acetate-HCl buffer (pH 5.5, adjusted with HCl), respectively, at 30 °C. After heat treatment at 100 °C for 5 min, the reaction mixtures (1 μL) were spotted onto TLC Silica Gel 60 F$_{254}$ (Merck, NJ, USA) plates. The plates were developed with 75% acetonitrile twice or more. The plates were then soaked in a 5% (w/v) sulfuric acid/methanol solution and heated in an oven until the spots were clearly visualized. A β-1,2-glucooligosaccharides marker was prepared by incubating Sop$_{3–7}$ and β-1,2-glucan in 1 mM sodium phosphate containing 1,2-β-oligoglucan phosphorylase from *Listeria innocua*[53].

**General properties.** To determine the optimum pH, EcOpgD (18.0 μg/ml) was incubated in various 20 mM buffers (sodium acetate-HCl, pH 4.0–5.5; Bis-Tris-HCl, pH 5.5–7.5; Tris-HCl, pH 7.5–9.0; glycine, pH 9.0–10.0) containing 0.8% β-1,2-glucan (treated with NaBH$_4$) at 30 °C for 10 min and then heated at 100 °C for 5 min to terminate the reaction. The reducing power of Sop$_n$s released from the substrate was measured by the MBTH method[52]. The optimum temperature was determined by performing the reactions in 20 mM sodium acetate buffer (pH 5.0) at each temperature (0–70 °C). The pH stability of EcOpgD was determined by incubating the purified enzyme (0.29 mg/mL) in various 20 mM buffers at 30 °C for an hour, and then the reaction was carried out in 50 mM sodium acetate buffer (pH 5.0) containing 0.8% β-1,2-glucan and the incubated EcOpgD (14.4 μg/mL as a final concentration) at 30 °C for 10 min. The temperature stability of EcOpgD was determined by incubating the enzyme (23.1 mg/mL) in 50 mM sodium acetate buffer (pH 5.0) at each temperature (0–70 °C) for an hour, and then the reaction was carried out under the same conditions as used for pH stability testing.

The optimum pH of EcOpgG (0.48 mg/mL) was determined by incubating the enzyme with 0.8% β-1,2-glucan (treated with NaBH$_4$) in various 50 mM buffers (sodium acetate-HCl, pH 4.0–5.5; MES-NaOH, pH 5.5–6.5; MOPS-NaOH, pH 6.5–7.5; Tris-HCl, pH 7.5–9.0; glycine, pH 9.0–10.0) at 30 °C for 15 min and then heated at 100 °C for 5 min to terminate the reaction. The reducing power of Sop$_n$s released from β-1,2-glucans was measured by the MBTH method[52].

**Substrate specificity.** EcOpgD (3.6 μg/mL) was incubated in 50 mM sodium acetate buffer (pH 5.0) containing each substrate (0.05% glucomannan, Neogen (Wicklow, Ireland); 0.0025% polygalacturonic acid, Neogen; 0.1% carboxymethyl cellulose, Merck; 0.02% soluble starch, Fujifilm; 0.005% carboxymethyl curdlan, Neogen; 0.0125% laminarin, Merck; 0.0125% lichenan, Neogen; 0.006% arabinogalactan, Neogen; 0.2% barley β-glucan, Neogen; 0.2% tamarind-xyloglucan, Neogen; 0.1% arabinan, Neogen; 0.05% pustulan, InvivoGen (CA, USA); or 0.8% β-1,2-glucan) at 30 °C for 10 min, and then the reducing power of products released from each substrate was measured by the MBTH method[52]. Glc was used as a standard for the assay using substrates, except for β-1,2-glucan. Sop$_2$ was used as a standard for the assay using β-1,2-glucans. EcOpgG (0.769 mg/mL) was incubated in 50 mM sodium acetate-HCl buffer (pH 5.5) containing 0.25% of each substrate (glucomannan, polygalacturonic acid, cellulose, starch, pachyman, laminarin, lichenan, arabinogalactan, barley β-glucan, tamarind-xyloglucan, arabinan, pustulan or β-1,2-glucan) at 37 °C for 24 h, and reaction patterns were then analyzed by TLC. The supernatants, after flush centrifugation of the reaction solutions for removal of insoluble substrates, were spotted on a TLC plate.

**Kinetic analysis.** The kinetic parameters for β-1,2-glucans were determined by performing the enzymatic reaction in a 20 μL reaction mixture containing 18.0 μg/mL EcOpgD, 0.0375–0.8 mM β-1,2-glucan (treated with NaBH$_4$) and 20 mM sodium acetate-HCl (pH 5.0) at 30 °C for 10 min. For EcOpgG, 0.48 mg/mL EcOpgG was used for the reaction in 50 mM sodium acetate-HCl (pH 5.5) for 15 min. Color development of the reaction mixtures was performed using the MBTH method[52]. Molar concentrations of β-1,2-glucans were calculated based on Mn. Kinetic parameters of EcOpgD were determined by fitting experimental data to the Michaelis–Menten equation, $v/[E]_0 = k_{cat} [S]/(K_m + [S])$, where $v$ is the initial velocity, $[E]_0$ is the enzyme concentration, $[S]$ is the substrate, $k_{cat}$ is the turnover number and $K_m$ is the Michaelis

constant. Each analysis was performed in triplicates and the medians were adopted.

**Polarimetric analysis of the reaction products of EcOpgD.** The time course of the degree of optical rotation in the reaction mixture was monitored to determine the reaction mechanism of EcOpgD. The degree of optical rotation of the reaction mixture containing EcOpgD (0.46 mg/mL) and 2% β-1,2-glucans was measured using a Jasco p1010 polarimeter (Jasco, Tokyo, Japan) at room temperature. Several drops of 25% aqueous ammonia were added 120 s after the reaction started to enhance mutarotation between the anomers.

In this method, during the enzymatic reaction phase, only either (α- or β-) anomer is produced enzymatically, resulting in change of an optical rotation of the solution derived from the newly produced anomer. After the reaction products are accumulated sufficiently, aqueous ammonia is added to accelerate nonenzymatic mutarotation of anomers, resulting in rapid change of optical rotations. In general, β-anomers have lower optical rotations than α-anomers, and the same is true for Sop$_n$s. If α-anomer is produced from a β-linked polymer by an inverting enzyme, the optical rotation quickly decreases after addition of aqueous ammonia due to increase of the ratio of β-anomer. Conversely, if β-anomer is produced by a retaining enzyme, the optical rotation quickly increases.

**Crystallography.** EcOpgD (D388N) and EcOpgG (D361N) were purified using a HisTrap™ FF crude column, as described above. The ligand-free crystals of EcOpgD for data collection were obtained at 20 °C after 3 days by mixing 1 μL D388N mutant (7.0 mg/mL) and 1 μL reservoir solution containing 0.1 M Tris-HCl buffer (pH 8.5), 0.1 M trimethylamine-N-oxide and 17% (w/v) polyethylene glycol (PEG) 2000 mono methyl ether. The crystals of EcOpgD for the β-1,2-glucan complex were obtained at 20 °C after several months by mixing 7 mg/mL of the D388N mutant with 1 μL 0.5% β-1,2-glucan and 1 μL reservoir solution comprising 0.1 M MMT (mixture of DL-malic acid, MES and Tris) (pH 4.0) and 20% (w/v) PEG 1500. The crystal of EcOpgG for the β-1,2-glucan complex was obtained at 20 °C after a day by mixing 1 μL of the D361N mutant (7 mg/mL) and 1 μL reservoir solution containing 0.1 M MMT (pH 5.0) and 27% (w/v) PEG 400. Ligand-free crystals of D388N were soaked in the reservoir solution supplemented with 27% (w/v) trehalose. The complex crystals of D388N were soaked in the reservoir solution supplemented with 22% (w/v) PEG 200 and 0.5% (w/v) β-1,2-glucan. The ligand-free crystals of D361N were soaked in a reservoir solution supplemented with 3% (w/v) β-1,2-glucan. Each crystal was kept at 100 K in a nitrogen-gas stream during data collection. All X-ray diffraction data were collected on a beamline (BL-5A) at Photon Factory (Tsukuba, Japan). The diffraction data for the ligand-free EcOpgD crystals, crystals of the β-1,2-glucan-bound EcOpgD and EcOpgG mutants were collected at 1.0 Å and processed with X-ray Detector Software (http://xds.mpimf-heidelberg.mpg.de/)[54] and the Aimless program (http://www.ccp4.ac.uk/). The initial phases of EcOpgD and EcOpgG structures were determined by molecular replacement using the Alphafold2 predicted EcOpgD and the ligand-free EcOpgG (PDB: 1txk) as model structures, respectively. Molecular replacement, auto model building and refinement were performed using MOLREP, Buccaneer, REFMAC5, and Coot programs, respectively (http://www.ccp4.ac.uk/)[55–58]. Crystallographic data collection and refinement statistics are summarized in Table 3. All visual representations of the structures were prepared using PyMOL (https://pymol.org/2/).

**Table 3 Crystallographic data collection and refinement statistics of EcOpgD and EcOpgG.**

| Data set | Ligand-free EcOpgD (D378N mutant) | EcOpgD-Sop$_{13}$ (D378N mutant) | EcOpgG-Sop$_{16}$ (D361N mutant) |
|---|---|---|---|
| *Data collection* | | | |
| Beamline | KEK BL-5A | KEK BL-5A | KEK BL-5A |
| Space group | C222 | P2$_1$ | C222$_1$ |
| Unit cell parameters (Å) | a = 226.62 | a = 58.09 | a = 62.82 |
| | b = 392.36 | b = 87.10 | b = 80.97 |
| | c = 324.48 | c = 110.85 | c = 213.23 |
| | | β = 101.13° | |
| Resolution (Å)[a] | 49.14-2.95 | 47.74-2.06 | 48.34-1.81 |
| | (3.00-2.95) | (2.11-2.06) | (1.85-1.81) |
| Total reflections[a] | 4,115,005 (209213) | 434,382 (21047) | 322,515 (19400) |
| Unique reflections[a] | 301,232 (14824) | 66,585 (4130) | 50,021 (2971) |
| Completeness (%)[a] | 100.0 (100.0) | 99.3 (92.6) | 99.9 (99.9) |
| Multiplicity[a] | 13.7 (14.1) | 6.5 (5.1) | 6.4 (6.5) |
| Mean I/σ(I)[a] | 13.0 (3.6) | 12.8 (2.1) | 17.2 (2.1) |
| $R_{merge}$ (%)[a] | 20.8 (87.8) | 11.6 (64.2) | 5.2 (76.2) |
| $R_{pim}$ (%)[a] | 8.4 (35.1) | 7.4 (46.6) | 3.3 (48.9) |
| $CC_{1/2}$[a] | (0.869) | (0.697) | (0.891) |
| *Refinement* | | | |
| Resolution (Å) | 49.14-2.95 | 47.74-2.06 | 48.34-1.81 |
| No. of reflections | 301,231 | 63,310 | 49,959 |
| No. of atoms | 51,848 | 9013 | 4118 |
| No. of water molecules | 666 | 566 | 122 |
| $R_{work}/R_{free}$ (%) | 19.4/23.2 | 17.5/22.4 | 21.3/25.4 |
| No. of asymmetric units | 12 | 2 | 1 |
| r.m.s.d. from ideal values | | | |
| Bond lengths (Å) | 0.0087 | 0.0084 | 0.014 |
| Bond angles (°) | 1.6025 | 1.5974 | 1.719 |
| Average B-factors (Å$^2$) | | | |
| Protein (chain A/B/C/...) | 34.7/40.3/37.9/35.1/35.3/37.9/34.6/40.5/37.9/35.2/34.7/40.5 | 24.1/24.4 | 45.1 |
| Ligand | | | |
| Sop$_n$s (chain A/B) | | 18.8/22.1 | 34.6 |
| Solvent | 32.0 | 25.8 | 34.4 |
| Ramachandran plot (%) | | | |
| Favored | 94.6 | 97.0 | 96.0 |
| Allowed | 5.2 | 3.0 | 4.0 |
| Outlier | 0.2 | 0.0 | 0 |
| PDB entry | 8IOX | 8IP1 | 8IP2 |

[a]Values in parentheses represent the highest resolution shell.

**Size-exclusion chromatography.** EcOpgD (0.5 mg/mL) and EcOpgG (0.5 mg/mL) were loaded onto a Superdex™ 200GL column (24 ml; Cytiva) equilibrated with 20 mM Tris-HCl buffer (pH 7.5) containing 100 mM NaCl, and then the target enzymes were eluted with the same buffer at the flow rate of 0.3 mL/min. Ovalbumin (44 kDa), conalbumin (75 kDa), aldolase (158 kDa), ferritin (440 kDa), and thyroglobulin (669 kDa) (Cytiva) were used as molecular weight markers. Blue dextran 2000 (2000 kDa) was used to determine the void volume of the column. The molecular weights of EcOpgD and EcOpgG were calculated using Eq. 1,

$$K_{av} = (V_e - V_o)/(V_t - V_o) \tag{1}$$

where $K_{av}$ is the gel-phase distribution coefficient; $V_e$ is the volume required to elute each protein; $V_o$ is the volume required to elute blue dextran 2000; and $V_t$ is the bed volume of the column.

**Mutational analysis.** The plasmids of EcOpgD and EcOpgG mutants were constructed using a PrimeSTAR mutagenesis basal kit (Takara Bio), according to the manufacturer's instructions. PCRs were performed using appropriate primer pairs (Supplemental Table S3) and the EcOpgD or EcOpgG plasmid as a template. Transformation into *E. coli* BL21(DE3) and the expression and purification of EcOpgD and EcOpgG mutants were performed using the same method described for wild-type EcOpgD preparation. The enzymatic reactions of EcOpgD mutants were performed basically in the same way as determining the optimum pH. The final assay concentrations of the mutants and reaction times were 0.0073–5.8 mg/mL and 0–2.5 h,

respectively, depending on the mutants. Color development was performed using the MBTH method[52].

**CD spectra**. The CD spectra analyses were performed at the range of 200–250 nm by using J820 (JASCO).

Each sample contained 2 mM Tris-HCl (pH 7.5) and each enzyme (0.0110–0.0125 mg/mL).

**Phylogenetic analysis**. ConSurf server[59,60] was used for visualizing conserved regions in the EcOpgD complex structure. Homologs that have 20–90% amino acid sequence identities with EcOpgD were comprehensively extracted by UNIREF-90, and 150 sequences were extracted evenly from those arranged in the order of homology. Homolog sequences collected in the same way using EcOpgG as a query were used for constructing the phylogenetic tree of the MdoG superfamily. The phylogenetic tree was prepared with the maximum likelihood method using MEGA11[61]. The structure-based multiple alignment was performed by PROMALS3D[62] and visualized using the ESPript 3.0 server (http://espript.ibcp.fr/ESPript/ESPript/)[63].

**Computational analysis**. Biding free energy between $Sop_{13}$ and protein was calculated by using scoring function of smina software[40]. The protein and ligand structure files were converted to pdbqt format by prepare_receptor4.py and prepare_ligand4.py script in MGLTools ver. 1.5.7[64].

For analysis of each residue contribution to substrate binding, the energy effect of a mutation on substrate binding affinity was calculated as the difference between the binding free energy in the mutated and wild-type protein ($\Delta\Delta G$) by Discovery Studio 2018 (BIOVIA, Dassault Systèmes, CA, USA). $\Delta\Delta G$ was defined as:

$$\Delta\Delta G = \Delta G_{mut} - \Delta G_{wild-type}$$

where $\Delta G_{mut}$ and $\Delta G_{wild-type}$ are the binding free energy of the mutant and wild-type enzyme, respectively. $Sop_{13}$ was used as the substrate. Residues of EcOpgD within 5 Å of the substrate in chain B were selected for analysis. The selected residues, except alanine, were substituted to alanine, whereas alanine was substituted to glycine.

**Statistics and reproducibility**. The quantitative data for hydrolytic activities of EcOpgD and EcOpgG were obtained from three independent experiments and medians were adopted. All multiplicity of obtained structures are more than 6.0.

**Reporting summary**. Further information on research design is available in the Nature Portfolio Reporting Summary linked to this article.

## Data availability

Atomic structure coordinates were deposited in the PDB under accession codes 8IOX, 8IP1, and 8IP2. The source data underlying Table 2, Fig. 1, Supplementary Figs. 1, 2, 9 and 13 are provided in Supplementary Data 1 (Supplementary Data 1.xlsx).

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

## Acknowledgements
This work was supported by Photon Factory for X-ray data collection (Proposal No. 2020G527). This work was supported in part by JST SPRING, Grant Number JPMJSP2151. We thank Edanz (https://jp.edanz.com/ac) for editing a draft of this manuscript.

## Author contributions
S.M., H.N., and M.N. conceived the project and designed the experiments. S.M. expressed and purified EcOpgG and EcOpgD. S.M. and M.N. performed data collection and processed X-ray crystallographic data. K.K. performed computational analysis. H.N. provided β-1,2-glucooligosaccharides. S.M., K.K., and M.N. prepared the manuscript. H.N. and M.N. supervised the project and participated in manuscript writing. All authors contributed to the revision of the manuscript. Any correspondence should be to M.N.

## Competing interests
The authors declare no competing interests.
