## [Peer Review File · Communications Biology]

Reviewers' comments:

Reviewer #1 (Remarks to the Author):

The study by Motouchi S. et al. reports the crystallographic structure and enzymatic characterisation of two proteins, OpgD and OpgG, from *Escherichia coli*, which are involved in the biosynthesis of osmoragulated periplasmic glucans (OPGs) and constitute a novel glycoside hydrolase (GH) family. OPGs are periplasmic or extracellular carbohydrates involved in the symbiotic and (phyto)pathogenic properties of several Gram-negative bacteria, such as *Escherichia coli*, *Salmonella Typhimurium* or *Pseudomonas aeruginosa*. The *opgGH* operon is responsible for the synthesis of OPGs, as *opgH* and/or *opgG* knockouts do not produce OPGs. While the X-ray structure of OpgG was solved about 20 years ago, its biological function and enzymatic activity remain elusive. In this study, the authors characterised the β -1,2-glucanase enzymatic activity of OpgD and OpgG. Although OpgG has the same substrate specificity (β -1,2-glucans), its activity is rather low (0.79% relative to OpgD). The authors then solved the X-ray structure of apo OpgD, a paralog of OpgG. They also solved the structure of both OpgD and OpgG in complex with a substrate (β -1,2-glucan). These new structures, in which OpgD and OpgG are organised as a homodimer, led the authors to decipher the substrate binding mode at the atomic level and to propose a catalytic mechanism involving an anomeric inversion mechanism. Once bound, the substrate is located in a tunnel formed by a cleft in the N-terminal domain of one chain and the LoopA (residues 434-453) of the other chain in the dimer. In OpgD, D388 represents a general acid catalyst and D300 a general base. A proton transfer mechanism is proposed from D300 to the nucleophilic water via a substrate hydroxyl group and two other water molecules. The length of the proton transfer pathway is unprecedented. LoopA contributes to this mechanism mainly through residues G440 and W441. In OpgG, the absence of the nucleophilic water molecule in the structure may explain its low enzymatic efficiency. This can be explained by the low sequence conservation for LoopA and the absence of hydrophobic residues in OpgG to trap this nucleophilic water. OpgD and OpgG belong to a new family of glycosylhydrolases.

The results are clear and informative. The manuscript is well written and overall interesting as it describes both the structure and the catalytic mechanism of a new family of glycosyl hydrolase (β -1,2-glucanase). These enzymes are involved in the biosynthesis of OPGs, which may mediate symbiotic or (phyto)pathogenic properties of Gram-negative bacteria.

Specific comments:

1. In complex with substrate, both OpgD and OpgG assemble as homodimers in the crystal packing. This is supported by the composite tunnel around the substrate, which consists of a cleft from the N-terminal domain of one chain and the loopA of the other chain. However, in the crystals of the apo forms, these proteins also assemble as homodimers and, to my personal knowledge, OpgG always behaves as a monomer in solution. It would have been valuable to study the monomeric/dimeric state of OpgD and OpgG in solution in the absence and presence of substrate.
2. While the enzymatic activity of OpgD is well characterised in the manuscript, this is less obvious for OpgG. Deciphering the exact enzymatic activity of OpgG is an important question. The mature OPGs in *E. coli* correspond to linear β -1,2-glucans (DP 5-12) with β -1,6 branches and various substitutions (phosphoglycerol, succinate, phosphoethanolamine). While the initial linear β -1,2-glucans, the addition of phosphoglycerol, succinate and/or phosphoethanolamine are performed by OpgH, OpgB, OpgC and OpgE, respectively, and the control of the length of the OPGs is performed by OpgD, as shown in this study, the enzyme responsible for the formation of the branches (β -1,6) remains to be identified. OpgG remains a possible candidate for this function. This is supported by the fact that the OpgD knockout mutant produces longer linear β -1,2-glucans with β -1,6 branches, whereas the OpgG

knockout mutant abolishes the synthesis of OPGs. However, it has been shown that the *opgG* and *opgD* genes are distributed widely, with many possible combinations; *opgG* alone, *opgG* and *opgD*, or *opgD* alone. The exact enzymatic function of *OpgG* is therefore still an open question that could have been further investigated. For example, it would have been interesting to characterise the mature OPGs synthesised in *E. coli* with a D361N or D273N mutation in *OpgG*.

Minor points:

- Page 6 line 149, it should be 'Fig. 2c right' instead of 'Fig. 2c left'.
- Page 7 line 162. It would be easier for the readers to explicitly state in the main text which water molecule, regarding to the nomenclature used in Figure 3, is discussed.
- Page 16 line 442, in the Methods section, Crystallography paragraph, it is not mentioned how much β -1,2 glucans were mixed with *OpgG* to get the complex.

Reviewer #2 (Remarks to the Author):

The manuscript describes the identification and characterisation of two beta-1,2-glucanases involved in the biosynthesis of osmo-regulated periplasmic glucans. Based on structural analysis and experimental mutational validation, the authors showed that one of the enzymes, *EcOpgD*, has a unique reaction mechanism, and belongs to a new GH family. The conclusions seem to be well-supported by the results from the experimental work.

The manuscript is well-written and easy to follow. I only have minor comments and suggestions to changes:

Lines 24+85+263+298: You say "report", I think it sounds better to refer to your work as "study"

I think that you have to introduce the CAZy family classification already in the introduction, instead of waiting until line 302 – maybe around lines 53-56, where you first mention a GH family.

Line 86: The abbreviation "SGL" for a beta-1,2-glucanase is not obvious to me. Is this a abbreviation used of historical reasons?

Line 96: Introduce what "Sopxx" stands for.

Line 178: It is not completely clear to me what you mean by "the subsite minus side from -5" – so you mean subsites -6, -7, and so on? If so, then just say subsites -5 to XX.

Line 216: Please give a reference to the Grotthuss mechanism.

Line 261-263: The sentence starting with "This is probably..." is not completely clear to me. This is not an explanation of why the enzymatic functions of *EcOpgD* and *EcOpgG* have not been investigated.

Line 296-297: Please give some references to knockout studies.
How did you determine protein concentration?

Lines 355-361: It is not clear to me if you prepared the beta-1,2-glucans yourself? If you did, then make it more clear, and if you did not, then tell where they were obtained from. What were their origin?

Line 372-380: What was the concentration of the enzymes used in the assay?

Line 382: Are the enzyme concentrations given in this section the final assay concentrations?

Line 408: It is not clear to me what you mean by the sentence "Glc was used as ...". Which molecule were you using as standard for beta-1,2-glucan?

Line 416 – kinetics analysis: An assay volume of 20 ul seems very low to me. However, I guess you are sure about evaporation not playing a role. Did you do the analysis in triplicates?

Line 439: What was the volume of the protein?

line 466: Is this final assay concentrations?

Figure 1C: I see no error bars on the graph, is it because the errors were so small that they cannot be seen, or did you only do the analysis in a single experiment?

Extended data Fig. 8: Where is EcOpgD+G found in the tree?

Maybe consider if the title of the manuscript could be improved? I guess that you want to point at the novel GH family, but in my opinion, the title is not clear.

Reviewer #3 (Remarks to the Author):

The manuscript by Motouchi et al represents a biochemical/structural tour-de-force to decipher the biochemical activity of OpgG and OpgD, two proteins required for the production of periplasmic glucans. These glucans have remained enigmatic in their physiological roles for a long time, and insight into the biochemistry of their generation is most welcome. I do not have any concerns about the data, with the caveat that this paper is written for a very specialized audience and I am unable to judge the structural data. The main task for the authors is to make this paper more accessible to a general audience, rather than a specialized circle of glycoside hydrolase experts.

1. In several places, there are sentences like "many pathogens lose their pathogenicity by knocking out opgG..." (e.g., in the abstract). This makes it sound like the bacterium itself deletes these genes to become less pathogenic (some sort of phase variation), but I am sure that is not what the authors mean?

2. Line 25/26 "with remarkably different activity" – different from what? Each other, or other hydrolases?

3. Line 96 – what is Sop?

4. Line 99 - what is SGL? I think this is never fully spelled out.

5. Line 116 and other places: They use a lot of jargon, which makes the paper very hard to read. Please make this more accessible to a general audience. What is optical rotation? What are "inverting SGLs" (line 120)? Skew boat conformations and Cremer-Pople parameters? I realize that not understanding this represents my own deficiency in this area, but Comms Bio is a journal for a more general audience, so please be mindful of this.

6. Line 125/126 – this statement could be supported with a structural alignment.

7. Line 146: How does obtaining a structure help you "delineate the very low reaction velocities"? Maybe rephrase to something like "to generate hypotheses regarding the very low reaction velocities...we determined the structure of..." to make it less strong. The structure itself does not

- delineate much, but the subsequent mutational and biochemical analysis (which is excellent), does.
8. Line 194 and following passage are impenetrable to non-experts. What is an "anomer inverting GH enzyme"? What does "all residues are via at least three water molecules"? (there might be a word missing in that phrase)?
 9. Line 200 how does low relative activity of the mutant suggest that Y356 is not a general base? I think I understand, but this could be spelled out better.
 10. Line 218 and following: What kind of data does this statement refer to? How did you determine that water molecules are sequestered? Is this based on the position of the water molecules in the structure or based on biochemistry?
 11. Line 248 – can you be certain that this has something to do with the catalytic center, rather than substrate binding? What are the K_d s of these enzymes?
 12. In general, make sure that all abbreviations are defined. SGL and GH are used in the discussion, and should be spelled out there once more. What does "DPs" stand for in Fig. 1 legend?

Reviewer #4 (Remarks to the Author):

Motouchi et al. characterized the recombinant enzymes of OpgG and its paralog OpgD, which had been suggested to be involved in osmo-regulated periplasmic glucan (OPG) biosynthesis in *Escherichia coli*, and found that they are β -1,2-glucanases. The kinetic parameters of OpgD toward β -1,2-glucan were within the range of those of known GHs, but those of OpgG for β -1,2-glucan were much lower. Although the ligand-free crystal structure of OpgG has been reported, the authors also succeeded in obtaining the Michaelis complex of the two enzymes, providing their substrate recognition mechanism and an unprecedented nucleophilic water activation mechanism mediated by two water molecules. They also obtained structural insights into the low activity of OpgG. They are founding members of a new GH family due to no homology to reported GH family enzymes and they employ the unique reaction mechanism, making this report extremely novel. OPG is an important carbohydrate not only in *E. coli* but also in various animal and plant pathogens, and its biosynthetic pathway is of interest to many microbiologists and infectious disease researchers. This manuscript is well written and the citations are generally appropriate, so I think it merits publication in this journal. However, several concerns need to be addressed before acceptance.

1. The similar assemblies of OpgD and OpgG in their crystal structure and the PISA analysis suggested that they form a dimer, but it would be better to provide other experimental evidence (e.g., size-exclusion chromatography and native PAGE). If the dimeric state has been reported previously, please describe it with a proper citation.
2. In line 109, the authors describe that the kinetic parameters of OpgD are still within the range of those of general GH enzymes. It would be helpful to have some more examples since readers who are not familiar with GHs do not know how much it is. The kinetic parameters of CMCase in the literature are those for the modified non-natural substrate (carboxymethyl cellulose), so it would be better to have examples of endo-acting GHs for soluble natural polysaccharides. CMCase is an abbreviation, so it should be endoglucanase or endo- β -1,4-glucanase.
3. Figures focusing on electron density maps for distorted ¹*S*₃ and ¹*S*₅ glucose molecules should be added because the distortion with the map is difficult to see from the viewpoint of Extended Data Figure 3. Since the Glc-1 in GH162, whose structure was analyzed by the authors' group, adopts 1S5 skew boat, please compare them and discuss the steric locations of anomeric carbon, glycosidic bond, nucleophilic water, and catalytic residues. Also, it would be helpful to mention other β -glucoside hydrolases that distort Glc-1 in ¹*S*₃ or ¹*S*₅ skew

boat.

4. The mutational analysis was well performed and the function of each amino acid residue is logically described. However, it is concerning that all the mutants were folded properly. The author should check them using CD spectra and so on.

Minor points:

line 51, cgs should be italics.

line 52, α -1,6-glucosidic is correct?

line 96, The abbreviation Sop_n (sophorooligosaccharide) should be defined here.

line 107, the first sequence-verified SGLs? The activity of SGLs was reported earlier.

line 158, in EcOpgD?

line 160 and others, S and C of ¹S₃, ¹S₅, and ⁴C₁ should be italics.

line 216, reference(s) of "the Grothuss mechanism in GH enzymes" should be cited here.

line 265, better to be β -1,6-glucosyl if referring to glucose only.

line 274, remove a space between Wat and 2.

line 294, Extended Data Fig. 9d?

line 335, provide a supplier name for PrimeSTAR Max.

line 337, SignalP5.0

lines 357-359, although using 'respectively', I cannot follow which average DP of glucan was used for each experiment. Please rewrite.

lines 438-443, spell out TMAO, PEG, MME, and MMT.

line 455, Are all program names in lowercase?

line 471, ConSurf

line 650 (Table 1 footnote b), please describe a unit of molar concentration (mM or M?).

line 651 (Table 1 footnote c), 0.8%(w/v) or 8 mg/mL?

Table 2, U should be defined.

Fig. 1b (top), N. D. is missing?

Fig. 1c, mg/ml should be mg/mL.

Extended Data Fig. 3, it would be better if both outermost subsite numbers are labeled.

Extended Data Fig. 8, it would be helpful if branches of EcOpgD and EcOpgG are marked.

Extended Data Fig. 9d, please add clade 14 (OpgG).

Supplementary Table 1, What does '(Subsite -7)' in the column of EcOpgG subsite -9 mean?

Similarly, for some other columns.

Responses to reviewers are written in red.

Reviewers' comments:

Reviewer #1 (Remarks to the Author):

The study by Motouchi S. et al. reports the crystallographic structure and enzymatic characterisation of two proteins, OpgD and OpgG, from *Escherichia coli*, which are involved in the biosynthesis of osmoragulated periplasmic glucans (OPGs) and constitute a novel glycoside hydrolase (GH) family. OPGs are periplasmic or extracellular carbohydrates involved in the symbiotic and (phyto)pathogenic properties of several Gram-negative bacteria, such as *Escherichia coli*, *Salmonella Typhimurium* or *Pseudomonas aeruginosa*. The *opgGH* operon is responsible for the synthesis of OPGs, as *opgH* and/or *opgG* knockouts do not produce OPGs. While the X-ray structure of OpgG was solved about 20 years ago, its biological function and enzymatic activity remain elusive. In this study, the authors characterised the β -1,2-glucanase enzymatic activity of OpgD and OpgG. Although OpgG has the same substrate specificity (β -1,2-glucans), its activity is rather low (0.79% relative to OpgD). The authors then solved the X-ray structure of apo OpgD, a paralog of OpgG. They also solved the structure of both OpgD and OpgG in complex with a substrate (β -1,2-glucan). These new structures, in which OpgD and OpgG are organised as a homodimer, led the authors to decipher the substrate binding mode at the atomic level and to propose a catalytic mechanism involving an anomeric inversion mechanism. Once bound, the substrate is located in a tunnel formed by a cleft in the N-terminal domain of one chain and the LoopA (residues 434-453) of the other chain in the dimer. In OpgD, D388 represents a general acid catalyst and D300 a general base. A proton transfer mechanism is proposed from D300 to the nucleophilic water via a substrate hydroxyl group and two other water molecules. The length of the proton transfer pathway is unprecedented. LoopA contributes to this mechanism mainly through residues G440 and W441. In OpgG, the absence of the nucleophilic water molecule in the structure may explain its low enzymatic efficiency. This can be explained by the low sequence conservation for LoopA and the absence of hydrophobic residues in OpgG to trap this nucleophilic water. OpgD and OpgG belong to a new family of glycosylhydrolases.

The results are clear and informative. The manuscript is well written and overall interesting as it describes both the structure and the catalytic mechanism of a new family of glycosyl hydrolase (β -1,2-glucanase). These enzymes are involved in the biosynthesis of OPGs, which

may mediate symbiotic or (phyto)pathogenic properties of Gram-negative bacteria.

Thank you for your positive comment and evaluation. We answered all your comments.

Specific comments:

1. In complex with substrate, both OpgD and OpgG assemble as homodimers in the crystal packing. This is supported by the composite tunnel around the substrate, which consists of a cleft from the N-terminal domain of one chain and the loopA of the other chain. However, in the crystals of the apo forms, these proteins also assemble as homodimers and, to my personal knowledge, OpgG always behaves as a monomer in solution. It would have been valuable to study the monomeric/dimeric state of OpgD and OpgG in solution in the absence and presence of substrate.

Answer

As you and referee #4 pointed out, size-exclusion chromatography was additionally performed for EcOpgD and EcOpgG, and both enzymes were found to assemble into dimers even as ligand-free forms. Supplemental Figure 1 was added and was referred at lines 134–135. In addition, although ref 26 describes that OpgG exists as a monomer in solution, this paper does not show any result nor describe method to evidence the assembly.

The binding dissociation energy of a dimer of EcOpgG calculated by PISA is different in the presence and absence of NaCl (13.5 and 3.0 kcal/mol, respectively). Thus, the assembly of EcOpgG may change depending on NaCl, although the dimer is still major assembly. In this revise, we performed SEC only in the presence of NaCl, because most of EcOpgG precipitated when concentration of NaCl decreased. This is described in Supplemental Discussion section.

2. While the enzymatic activity of OpgD is well characterised in the manuscript, this is less obvious for OpgG. Deciphering the exact enzymatic activity of OpgG is an important question.

The mature OPGs in *E. coli* correspond to linear β -1,2-glucans (DP 5–12) with β -1,6 branches and various substitutions (phosphoglycerol, succinate, phosphoethanolamine). While the initial linear β -1,2-glucans, the addition of phosphoglycerol, succinate and/or phosphoethanolamine are performed by OpgH, OggB, OpgC and OpgE, respectively, and the control of the length of the OPGs is performed by OpgD, as shown in this study, the enzyme responsible for the formation of the branches (β -1,6) remains to be identified. OpgG remains a possible candidate for this function. This is supported by the fact that the OpgD knockout mutant produces longer linear β -1,2-glucans with β -1,6 branches, whereas the OpgG knockout mutant abolishes the synthesis of OPGs. However, it has been shown that the *opgG* and *opgD* genes are distributed widely, with many possible combinations; *opgG* alone, *opgG* and *opgD*, or *opgD* alone. The exact enzymatic function of OpgG is therefore still an open question that could have been further investigated. For example, it would have been interesting to characterise the mature OPGs synthesised in *E. coli* with a D361N or D273N mutation in OpgG.

Answer

As you point out, the true function of EcOpgG remains unknown. Since that point is of interest to us, we will continue our work on EcOpgG to biochemically identify its function later. We are planning to address phenotypes of the catalytic residue mutant strains of EcOpgD/EcOpgG in our future papers.

Aside from our plans, speculation of EcOpgG functions and destiny of OPG are further discussed at lines 297–303.

“The possibility that EcOpgG is in fact a transglycosylase of the β -1,6-glucose side chain remains open. Knockout mutants of OpgG are supposed not to synthesize OPG precursors with β -1,2-glucan main chain. However, this may simply because the precursors are not detected. *E. coli* has only one β -glucosidase homolog predicted to be localized in the periplasmic space. The β -glucosidase from *E. coli* is a close homolog of β -glucosidases specific to Sop_ns as native substrates^{33,41}, and the recognition residues for subsite +1 in the homolog are conserved in the *E. coli* β -glucosidase. Therefore, if knockout of *opgG* results in just synthesizing unbranched β -1,2-glucans, they are likely to be degraded.”

Minor points:

1. – Page 6 line 149, it should be ‘Fig. 2c right’ instead of ‘Fig. 2c left’.

The label was revised as instructed (line 161, revised position).

2. – Page 7 line 162. It would be easier for the readers to explicitly state in the main text which water molecule, regarding to the nomenclature used in Figure 3, is discussed.

(Wat1) was added as instructed (line 178, revised position).

3. – Page 16 line 442, in the Methods section, Crystallography paragraph, it is not mentioned how much β -1,2 glucans were mixed with OpgG to get the complex.

Firstly, the crystal was made as ligand-free crystal, and soaked in 3% β -1,2-glucan when the data was collected (lines 486).

Reviewer #2 (Remarks to the Author):

The manuscript describes the identification and characterisation of two beta-1,2-glucanases involved in the biosynthesis of osmo-regulated periplasmic glucans. Based on structural analysis and experimental mutational validation, the authors showed that one of the enzymes, EcOpgD, has a unique reaction mechanism, and belongs to a new GH family. The conclusions seem to be well-supported by the results from the experimental work. The manuscript is well-written and easy to follow. I only have minor comments and suggestions to changes:

Thanks for positive comments and for listing the additions needed to more accurately convey what we want to communicate. We answered all your comments.

Minor comments

1. Lines 24+85+263+298: You say “report”, I think it sounds better to refer to your work as “study”

All “report” is replaced with “study” as instructed (lines 24, 85, 283 and 324, revised positions).

2. I think that you have to introduce the CAZy family classification already in the introduction, instead of waiting until line 302 – maybe around lines 53–56, where you first mention a GH family.

Brief description on the CAZy database was added at lines 55–56 as you pointed out.

3. Line 86: The abbreviation “SGL” for a beta-1,2-glucanase is not obvious to me. Is this a abbreviation used of historical reasons?

If the abbreviation for beta-glucanase is abbreviated to BGL, it is indistinguishable from Beta-glucosidase. In addition, there seems not to be a defined abbreviation for beta-glucanase.

Therefore, when we first published the paper of *C. pinensis* beta-1,2-glucanase, we came up with an acronym that would be easy to distinguish. “S” stands for sophoro (-oligosaccharide), an alternative name of b-1,2-gluco-oligosaccharide. To show this point clearly, the origin of “S” was added to the text (lines 86–87).

4. Line 96: Introduce what “Sopxx” stands for.

“(Sopn: sophorooligosaccharide with DP of n)” was added after ”Sop6–7” as instructed (line 97, revised position).

5. Line 178: It is not completely clear to me what you mean by “the subsite minus side from –5” – so you mean subsites –6, –7, and so on? If so, then just say subsites –5 to XX.

The point was revised as instructed (line 194, revised position).

6. Line 216: Please give a reference to the Grotthuss mechanism.

The papers for GH6 andGH136 were cited as instructed (lines 235–236, revised positions).

7. Line 261–263: The sentence starting with “This is probably…” is not completely clear to me. This is not an explanation of why the enzymatic functions of EcOpgD and EcOpgG have not been investigated.

As you pointed out, the sentence did not explain why the enzymatic functions of EcOpgD and EcOpgG have not been investigated. Therefore, the sentence was deleted.

8. Line 296–297: Please give some references to knockout studies.

Reference was added as instructed (line 325, revised position).

9. How did you determine protein concentration?

Absorbance at 280 nm was used and the method was added at lines 382.

10. Lines 355–361: It is not clear to me if you prepared the beta-1,2-glucans yourself? If you did, then make it more clear, and if you did not, then tell where they were obtained from. What were their origin?

We prepared the beta-1,2-glucan by ourselves. It was referred at the last of the section (lines 390–391, revised positions).

11. Line 372–380: What was the concentration of the enzymes used in the assay?

Final concentration of EcOpgD and EcOpgG were respectively added in parenthesis as instructed (line 403, revised position).

12. Line 382: Are the enzyme concentrations given in this section the final assay concentrations?

0.29 mg/mL is not final concentration. Final concentration for pH stability is 14.4 ug/mL.. Other concentrations indicate final concentrations. Therefore, the final concentration of EcOpgD was added at the sentence of pH stability (line 421, revised position).

13. Line 408: It is not clear to me what you mean by the sentence “Glc was used as …”. Which molecule were you using as standard for beta-1,2-glucan?

The standard used for β -1,2-glucan is Sop2 and description for the content was added (line 440, revised position)

14. Line 416 – kinetics analysis: An assay volume of 20 ul seems very low to me. However, I guess you are sure about evaporation not playing a role. Did you do the analysis in triplicates?

As you point out, the experiment was conducted in triplicates. And the median was adopted. We added the sentence and corresponding graphs were revised (line 456 Fig. 1c, revised position). These assays were performed using a PCR equipment and evaporation was kept minimized. This could be added in the text if needed. Assay volume was small to save the amounts of self-prepared substrates.

15. Line 439: What was the volume of the protein?

D388N mutant (of EcOpgD) was used, and the volume was described as 1 uL (line 476).

16. Line 466: Is this final assay concentrations?

Yes. “final assay” was added (line 520, revised position)

17. Figure 1C: I see no error bars on the graph, is it because the errors were so small that they cannot be seen, or did you only do the analysis in a single experiment?

We performed kinetic analyses in triplicates. Therefore, error bars were added as instructed (Fig. 1c).

18. Extended data Fig. 8: Where is EcOpgD+G found in the tree?

The names of EcOpgD and EcOpgG were added in clade2 and clade14 respectively as instructed. The position of each branch was indicated by a red arrow.

19. Maybe consider if the title of the manuscript could be improved? I guess that you want to point at the novel GH family, but in my opinion, the title is not clear.

The title was changed to “Structural and functional analyses of osmo-regulated periplasmic glucans biosynthesis proteins, OpgG and OpgD from Escherichia coli reveal a novel glycoside hydrolase family”, based on your suggestion.

Reviewer #3 (Remarks to the Author):

The manuscript by Motouchi et al represents a biochemical/structural tour-de-force to decipher the biochemical activity of OpgG and OpgD, two proteins required for the production of periplasmic glucans. These glucans have remained enigmatic in their physiological roles for a long time, and insight into the biochemistry of their generation is most welcome. I do not have any concerns about the data, with the caveat that this paper is written for a very specialized audience and I am unable to judge the structural data. The main task for the authors is to make this paper more accessible to a general audience, rather than a specialized circle of glycoside hydrolase experts.

Thank you for clarifying the points that need to be explained in detail in order to deliver our outcome to a wide audience. We answered all your comments.

1. In several places, there are sentences like “many pathogens lose their pathogenicity by knocking out *opgG*...” (e.g., in the abstract). This makes it sound like the bacterium itself deletes these genes to become less pathogenic (some sort of phase variation), but I am sure that is not what the authors mean?

As you pointed out, those statements were intended to indicate that the pathogen would lose its virulence due to artificial mutations. Therefore, we have rewritten the related statements (abstract, lines 22 and 40, revised positions).

2. Line 25/26 “with remarkably different activity” – different from what? Each other, or other hydrolases?

“from each other” was added as pointed out (line 26, revised position).

3. Line 96 – what is Sop?

A definition of the abbreviation was placed after “Sop6–7” as instructed (line 97, revised position).

4. Line 99 – what is SGL? I think this is never fully spelled out.

As you pointed out, the abbreviation was defined. S stands for Sophoro (–oligosaccharide). GL stands for GLucanase (lines 86– 87, revised position). The reason why “S” is used in the abbreviation is briefly referred.

5. Line 116 and other places: They use a lot of jargon, which makes the paper very hard to read. Please make this more accessible to a general audience. What is optical rotation? What are “inverting SGLs” (line 120)? Skew boat conformations and Cremer–Pople parameters? I realize that not understanding this represents my own deficiency in this area, but Comms Bio is a journal for a more general audience, so please be mindful of this.

The anomers at the reducing end of the hydrolysis product were examined to determine whether they were inverted or retained by measuring the change in optical rotation of the enzyme reaction solution. As you pointed out, this text is highly technical and difficult to understand on first reading, so we included a detailed explanation of why we performed this experiment (lines 119–125, revised positions) and the principle in the main text (lines 464–471, revised position). This description also helped the reader to understand the meaning of the anomer–inverting and anomer–retaining types.

The skew boat conformation of the 6–membered ring is classified as a “distorted conformation” and we added that to the text (line 174, revised position).

Cremer–Pople parameters is the puckering coordinate system of a six–membered pyranose ring. Sugar packings are broadly classified into chair type, half chair type, boat type, and skew boat type. Based on the Cremer–pople parameters, we can determine the type of conformation of the monosaccharide of interest. Therefore, “the puckering coordinate system of a six–membered pyranose ring” was added (lines 176–177, revised positions)

6. Line 125/126 – this statement could be supported with a structural alignment.

The RMSD value between ligand-free EcOpgD and EcOpgG, which was listed in the legend of Fig. 2 (lines 759–760), was also included in the main text as you pointed out (lines 137–138, revised position). A diagram showing the superposition of two ligand-free structures was also added as supplemental figure 2.

7. Line 146: How does obtaining a structure help you “delineate the very low reaction velocities”? Maybe rephrase to something like “to generate hypotheses regarding the very low reaction velocities…we determined the structure of…” to make it less strong. The structure itself does not delineate much, but the subsequent mutational and biochemical analysis (which is excellent), does.

Since your proposed alternative sentence is based on more correct logic than the original sentence, I adopted that proposal (lines 157–158, revised positions).

8. Line 194 and following passage are impenetrable to non-experts. What is an “anomer inverting GH enzyme”? What does “all residues are via at least three water molecules”? (there might be a word missing in that phrase)?

As you pointed out, the explanation for “Anomer-inverting” was added as above.

The sentence including “all residues are via at least three water molecules” was changed to “All listed residues need at least three water molecules, including the nucleophilic water, in order to transfer the proton from nucleophilic water to each residue.” (lines 215–216, revised positions).

9. Line 200 how does low relative activity of the mutant suggest that Y356 is not a general base? I think I understand, but this could be spelled out better.

We thought “14% “means “not fully decreased”. So, added as that (line 219, revised position).

10. Line 218 and following: What kind of data does this statement refer to? How did you determine that water molecules are sequestered? Is this based on the position of the water molecules in the structure or based on biochemistry?

This is based on the position of the water molecules in the structure. “Structurally” was added Fig. 4 was referenced as we intended (line 238, revised position).

11. Line 248 – can you be certain that this has something to do with the catalytic center, rather than substrate binding? What are the Kds of these enzymes?

Each binding free energy between Sop13 and the enzymes was additionally calculated. The values of EcOpgD and EcOpgG are -6.33526 kcal/mol and -5.87115 kcal/mol respectively. This means each binding free energy is almost the same each other and explains that the low hydrolytic activity of EcOpgG is not caused by substrate binding (lines 267–270, revised positions)

12. In general, make sure that all abbreviations are defined. SGL and GH are used in the discussion, and should be spelled out there once more. What does “DPs” stand for in Fig. 1 legend?

As you pointed out, SGL and GH are spelled out in discussion again (lines 283 and 295, revised positions). DPs are spelled out in introduction (line 68).

Reviewer #4 (Remarks to the Author):

Motouchi et al. characterized the recombinant enzymes of OpgG and its paralog OpgD, which had been suggested to be involved in osmo-regulated periplasmic glucan (OPG) biosynthesis in *Escherichia coli*, and found that they are β -1,2-glucanases. The kinetic parameters of OpgD toward β -1,2-glucan were within the range of those of known GHs, but those of OpgG for β -1,2-glucan were much lower. Although the ligand-free crystal structure of OpgG has been reported, the authors also succeeded in obtaining the Michaelis complex of the two enzymes, providing their substrate recognition mechanism and an unprecedented nucleophilic water activation mechanism mediated by two water molecules. They also obtained structural insights into the low activity of OpgG. They are founding members of a new GH family due to no homology to reported GH family enzymes and they employ the unique reaction mechanism, making this report extremely novel. OPG is an important carbohydrate not only in *E. coli* but also in various animal and plant pathogens, and its biosynthetic pathway is of interest to many microbiologists and infectious disease researchers. This manuscript is well written and the citations are generally appropriate, so I think it merits publication in this journal. However, several concerns need to be addressed before acceptance.

Thank you for your positive comments and for providing specific advice that make this paper more relevant in the eyes of CAzymes professionals. We answered all your comments.

1. The similar assemblies of OpgD and OpgG in their crystal structure and the PISA analysis suggested that they form a dimer, but it would be better to provide other experimental evidence (e.g., size-exclusion chromatography and native PAGE). If the dimeric state has been reported previously, please describe it with a proper citation.

As you pointed out, size-exclusion chromatography was additionally performed for EcOpgD and EcOpgG, and both enzymes were found to assemble into a dimer even in a ligand-free form. The supplemental figure 1 was added and was referred at lines 134 and 135.

2. In line 109, the authors describe that the kinetic parameters of OpgD are still within the range of those of general GH enzymes. It would be helpful to have some more examples since readers who are not familiar with GHs do not know how much it is. The kinetic parameters of CMCase in the literature are those for the modified non-natural substrate (carboxymethyl cellulose), so it would be better to have examples of endo-acting GHs for soluble natural polysaccharides. CMCase is an abbreviation, so it should be endoglucanase or endo- β -1,4-glucanase.

As you pointed out, alternative reference was cited, and main text was revised as that (lines 112–114, revised positions).

As far as I know, there are many endo-glucanases that show large activity with a single digit of polymerization of the original substrate, and there are not many papers that calculate kinetics with substrates with a degree of polymerization greater than 100. Here, we cite a paper on dextranase, an endo- α -glucanase, as one of the few examples. The K_m of this enzyme for dextran with a degree of polymerization of about 2000 is about 5.8 mg/mL, which is larger than that of EcOpgD for its native soluble substrate, so it was cited for comparison at lines 112–114.

3. Figures focusing on electron density maps for distorted 1S3 and 1S5 glucose molecules should be added because the distortion with the map is difficult to see from the viewpoint of Extended Data Figure 3. Since the Glc-1 in GH162, whose structure was analyzed by the authors' group, adopts 1S5 skew boat, please compare them and discuss the steric locations of anomeric carbon, glycosidic bond, nucleophilic water, and catalytic residues. Also, it would be helpful to mention other β -glucoside hydrolases that distort Glc-1 in 1S3 or 1S5 skew boat.

As instructed, supplemental figure 3 was added, and cited at lines 176 and 184. The 1S3 skew boat Glc moiety of GH162 and GH3 was referenced as an example of distorted Glc moiety at scissile bond, as instructed.

4. The mutational analysis was well performed and the function of each amino acid residue is logically described. However, it is concerning that all the mutants were folded properly. The author should check them using CD spectra and so on.

As you pointed out, CD spectra was measured for the mutants whose structure are not obtained. All examined mutants showed similar CD spectra patterns with WT EcOpgD (Supplemental figure 4, revised position).

Minor points:

line 51, cgs should be italics.

Revised as instructed (line 51, revised position).

line 52, α -1,6-glucosidic is correct?

Yes (lines 45-46). \$\alpha\$ -1,6-cyclized \$\beta\$ -1,2-glucan was extracted directly from *Xanthomonas campestris* and its structure was solved, but no synthetic enzyme has been identified.

line 96, The abbreviation Sopn (sophorooligosaccharide) should be defined here.

Sopn was defined as instructed (line 97, revised position).

line 107, the first sequence-verified SGLs? The activity of SGLs was reported earlier.

As you pointed out, “sequence-verified” was added (line 109, revised position).

line 158, in EcOpgD?

Yes. “in EcOpgD” was added at line 171.

line 160 and others, S and C of 1S3, 1S5, and 4C1 should be italics.

Revised as instructed (lines 173 and 175 and 177 and 197, revised positions).

line 216, reference(s) of “the Grotthuss mechanism in GH enzymes” should be cited here.

The study for GH6 and 136 were cited as instructed (lines 235–236, revised positions).

line 265, better to be β -1,6-glucosyl if referring to glucose only.

Revised as instructed (line 285, revised position).

line 274, remove a space between Wat and 2.

Revised as instructed (line 294, revised position).

line 294, Extended Data Fig. 9d?

It was corrected as you pointed out (line 322, revised position).

line 335, provide a supplier name for PrimeSTAR Max.

As you pointed out, the supplier was added (lines 363–364, revised positions).

line 337, SignalP5.0

Revised as instructed (line 365, revised position).

lines 357–359, although using ‘respectively’, I cannot follow which average DP of glucan was used for each experiment. Please rewrite.

As you pointed out, the sentences were rewritten (lines 386–389, revised positions).

lines 438–443, spell out TMAO, PEG, MME, and MMT.

As you pointed out, these abbreviations were spelled out (lines 477–478 and 480, revised positions).

line 455, Are all program names in lowercase?

These names were revised as instructed (line 495, revised position).

line 471, ConSurf

Revised as instructed (line 529).

line 650 (Table 1 footnote b), please describe a unit of molar concentration (mM or M?).

“(mM)” was added as instructed (line 726, revised position).

line 651 (Table 1 footnote c), 0.8%(w/v) or 8 mg/mL?

“8 mg/mL” was added as pointed out (line 727, revised position).

Table 2, U should be defined.

As you pointed out, U was defined at line 732.

Fig. 1b (top), N. D. is missing?

As you pointed out, N. D. is added on the figure.

Fig. 1c, mg/ml should be mg/mL.

Revised as instructed.

Extended Data Fig. 3, it would be better if both outermost subsite numbers are labeled.

Revised as instructed.

Extended Data Fig. 8, it would be helpful if branches of EcOpgD and EcOpgG are marked.

Marked as instructed.

Extended Data Fig. 9d, please add clade 14 (OpgG).

Added as instructed.

Supplementary Table 1, What does '(Subsite -7)' in the column of EcOpgG subsite -9 mean? Similarly, for some other columns.

The explanation was added at last sentence of legend of Supplemental Table 1, as you pointed out.

REVIEWERS' COMMENTS:

Reviewer #1 (Remarks to the Author):

Overall, the quality of the revised manuscript has been further improved and all points raised have been accordingly addressed by the authors.

Reviewer #2 (Remarks to the Author):

The authors have replied carefully to all my comments. Furthermore, I think that they have also addressed the very insight full comments from the other reviewers. I do not have any further comments.

Reviewer #3 (Remarks to the Author):

My comments have been addressed and I have no further concerns.

Reviewer #4 (Remarks to the Author):

Motouchi et al. characterized the recombinant enzymes of OpgG and its paralog OpgD, which had been suggested to be involved in osmo-regulated periplasmic glucan (OPG) biosynthesis in *Escherichia coli*, and found that they are β -1,2-glucanases. The kinetic parameters of OpgD toward β -1,2-glucan were within the range of those of known GHs, but those of OpgG for β -1,2-glucan were much lower. Although the ligand-free crystal structure of OpgG has been reported, the authors also succeeded in obtaining the Michaelis complex of the two enzymes, providing their substrate recognition mechanism and an unprecedented nucleophilic water activation mechanism mediated by two water molecules. They also obtained structural insights into the low activity of OpgG. They are founding members of a new GH family due to no homology to reported GH family enzymes and they employ the unique reaction mechanism, making this report extremely novel. OPG is an important carbohydrate not only in *E. coli* but also in various animal and plant pathogens, and its biosynthetic pathway is of interest to many microbiologists and infectious disease researchers. The authors did nice work, this manuscript is totally well written and my concerns have been properly addressed. I recommend acceptance of the manuscript after addressing grammar issues newly raised in the revised manuscript.

Title, reveal  reveals (The subject is 'identification')

Line 40, "are lost"  "is lost" (The subject is 'Pathogenicity')

Line 123, "during the reaction the same way"  "during the reaction in the same way"

Line 194, "at the subsite -5 to -7"  "at the subsites -5 to -7"

Lines 235 and 236, "GH enzymes (GH6, GH136)"  "GH enzymes (GH6 and GH136)"

Line 299, "this may simply because"  "this may simply be because"

Line 537, "Biding free energy between {Sop13 and protein} was"  "Binding free energy between Sop13 and protein was"

Responses to reviewers are written in red.

REVIEWERS' COMMENTS:

Reviewer #1 (Remarks to the Author):

Overall, the quality of the revised manuscript has been further improved and all points raised have been accordingly addressed by the authors.

Reviewer #2 (Remarks to the Author):

The authors have replied carefully to all my comments. Furthermore, I think that they have also addressed the very insight full comments from the other reviewers.

I do not have any further comments.

Reviewer #3 (Remarks to the Author):

My comments have been addressed and I have no further concerns.

Reviewer #4 (Remarks to the Author):

Motouchi et al. characterized the recombinant enzymes of OpgG and its paralog OpgD, which had been suggested to be involved in osmo-regulated periplasmic glucan (OPG) biosynthesis in *Escherichia coli*, and found that they are β -1,2-glucanases. The kinetic parameters of OpgD toward β -1,2-glucan were within the range of those of known GHs, but those of OpgG for β -1,2-glucan were much lower. Although the ligand-free crystal structure of OpgG has been reported, the authors also succeeded in obtaining the Michaelis complex of the two enzymes, providing their substrate recognition mechanism and an unprecedented nucleophilic water activation mechanism mediated by two water molecules. They also obtained structural insights into the low activity of OpgG. They are founding members of a new GH family due to no homology to reported GH family enzymes and they employ the unique reaction mechanism, making this report extremely novel. OPG is an important carbohydrate not only in *E. coli* but also in various animal and plant pathogens, and its biosynthetic pathway is of interest to many microbiologists and infectious disease researchers. The authors did nice work, this manuscript is totally well written and my concerns have been properly addressed. I recommend acceptance of the manuscript after addressing grammar issues newly raised in the revised manuscript.

Thank you for your additional careful comments.

All points were revised as instructed.

Title, reveal  reveals (The subject is 'identification')

Line 40, "are lost"  "is lost" (The subject is 'Pathogenicity')

(Revised version line 42)

Line 123, "during the reaction the same way"  "during the reaction in the same way"

(Revised version line 125)

Line 194, "at the subsite -5 to -7"  "at the subsites -5 to -7"

(Revised version line 196)

Lines 235 and 236, "GH enzymes (GH6, GH136)"  "GH enzymes (GH6 and GH136)"

(Revised version line 237)

Line 299, "this may simply because"  "this may simply be because"

(Revised version line 301)

Line 537, "Biding free energy between {Sop13 and protein} was"  "Binding free energy between Sop13 and protein was"

(Revised version line 538)

Other correction

Typo: Table 2 T386S 17.5 to 17.6 (%)

Addition: Supplementary Table 2 Primers used in this study.

Extended data are combined with the supplementary information

Corrections related to the check list for final revision instructions.